# DA²: Depth Anything in Any Direction

**Haodong Li**[123§], **Wangguandong Zheng**[1], **Jing He**[3], **Yuhao Liu**[1], **Xin Lin**[2], **Xin Yang**[34],
**Ying-Cong Chen**[34‡], **Chunchao Guo**[1‡]
[1]Tencent Hunyuan [2]UC San Diego [3]HKUST(GZ) [4]HKUST
hal211@ucsd.edu; yingcongchen@ust.hk; chunchaoguo@gmail.com

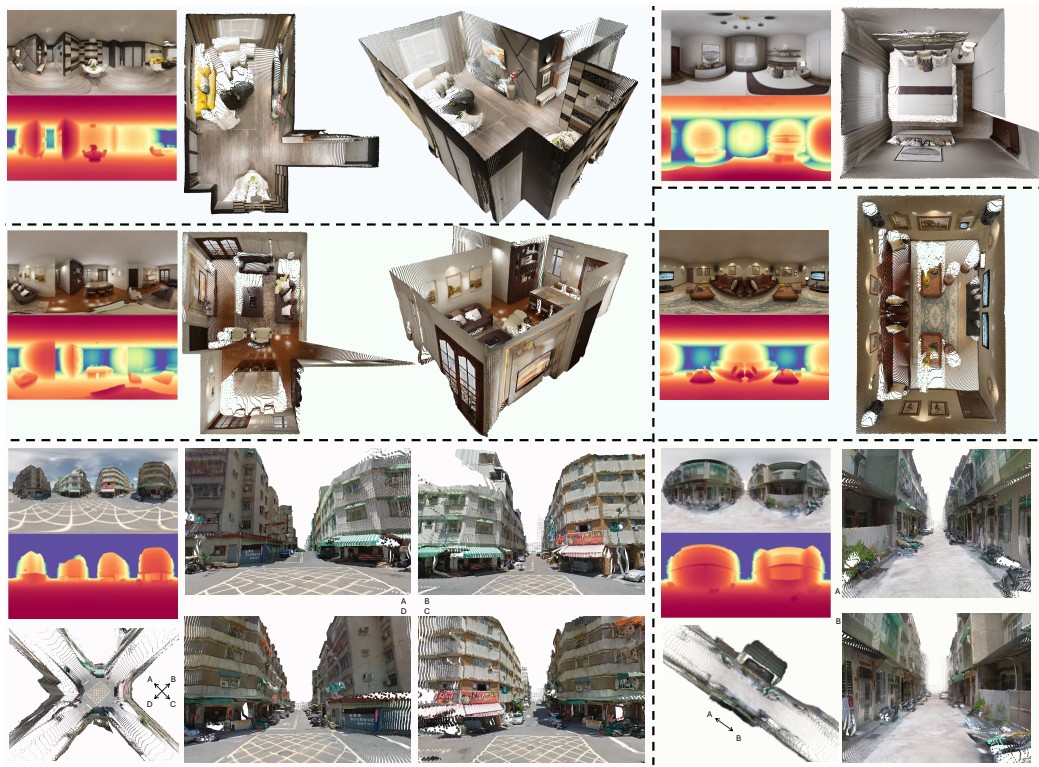

Figure 1: Teaser of DA². Powered by large-scale training data from our panoramic data curation engine, and the distortion-aware SphereViT, DA² predicts dense distance from a single 360° panorama, with remarkable geometric fidelity. The reconstructed 3D structures exhibit sharp geometric details and robust performance across diverse scenes, highlighting DA²'s strong zero-shot generalization.

## ABSTRACT

Panorama has a full FoV (360°×180°), offering a more complete visual description than perspective images. Thanks to this characteristic, panoramic depth estimation is gaining increasing traction in 3D vision. However, due to the scarcity of panoramic data, previous methods are often restricted to in-domain settings, leading to poor zero-shot generalization. Furthermore, due to the spherical distortions inherent in panoramas, many approaches rely on perspective splitting (*e.g.*, cubemaps), which leads to suboptimal efficiency. To address these challenges, we propose **DA²: D**epth **A**nything in **A**ny **D**irection, an accurate, zero-shot generalizable, and fully end-to-end panoramic depth estimator. Specifically, for scaling up panoramic data, we introduce a data curation engine for generating high-quality panoramic depth data from perspective, and create ∼543K panoramic RGB-depth pairs, bringing the total to ∼607K. To further mitigate the spherical distortions, we present SphereViT, which explicitly leverages spherical coordinates to enforce the spherical geometric consistency in panoramic image features, yielding improved performance. A comprehensive benchmark on multiple datasets clearly demonstrates DA²'s SoTA performance, with an average 38% improvement on AbsRel over the strongest zero-shot baseline. Surprisingly, DA² even outperforms prior in-domain methods, highlighting its superior zero-shot generalization. Moreover,

as an end-to-end solution, $DA^2$ exhibits much higher efficiency over fusion-based approaches. Both the code and the curated panoramic data have be released. Project page: `https://depth-any-in-any-dir.github.io/`

# 1 INTRODUCTION

Unlike the commonly used perspective images, panorama offers an immersive $360° \times 180°$ view, capturing visual content from *any direction*. This wide FoV makes panorama an essential visual representation in computer vision, empowering a variety of exciting applications, such as AR/VR (Chen et al., 2023) and immersive visual generation (Yang et al., 2025a; Kalischek et al., 2025). However, immersive visual (2D) experiences alone are not enough. To push the new frontier of panoramic application scenarios, high-quality depth (3D) information from panoramas is crucially needed for 3D reconstruction and more advanced features such as 3D scene generation (Skywork AI, 2025; Li et al., 2025; Lu et al., 2025), physical simulation (Shah et al., 2025), etc. Inspired by this, we focuses on estimating scale-invariant[1] distance[2] from each panorama pixel to the sphere center (*i.e.,* the $360°$ camera) in an end-to-end manner, with high-fidelity and strong zero-shot generalization.

Panoramic depth estimation is particularly valuable for applications requiring comprehensive spatial awareness. However, capturing or rendering panoramas is much more challenging than perspective images, panoramic depth data is much more limited in both quantity and diversity. Consequently, early methods were largely trained and tested in in-domain settings, with highly limited zero-shot generalization. Given the wealth of high-quality perspective depth data, is it possible to transform them into panoramic? Motivated by this, we propose a data curation engine, transforming perspective samples into high-quality panoramic data. Concretely, given a perspective RGB image with known horizontal and vertical FoVs, we first apply Perspective-to-Equirectangular (P2E) projection to map the image onto the spherical space. However, due to the limited FoV of perspective images (with a typical horizontal range of $70°-90°$), only a small portion of the spherical space can be covered (as highlighted in Fig. 3's left sphere). Thus, such a P2E projected image can be viewed as an "incomplete" panorama. Then, panoramic out-painting will be performed to generate a "complete" panorama to match the input of our model, using an image-to-panorama out-painter: FLUX-I2P (BFL, 2024; Tencent, 2025). For the associated GT depth, we apply only the P2E projection *without* out-painting, due to concerns on the *absolute accuracy* of out-painted depth. Overall, this data curation engine substantially boosts the quantity and diversity of panoramic data, and significantly strengthens the zero-shot performance of $DA^2$, as shown in Fig. 2 and Tab. 2.

Panoramas typically use equirectangular projection (ERP)[3] to represent the $360° \times 180°$ visual space. However, a 3D spherical space cannot be "losslessly" projected onto a 2D plane. During the sphere-to-plane projection, distortions and stretching are inevitable, particularly near the poles. This spherical distortion is analogous to the challenge in world map projection, where you can never accurately express both the areas and shapes of each land. To mitigate the impact of spherical distortion, inspired by the positional embeddings in Vision Transformers (ViTs), we propose SphereViT—the main backbone of $DA^2$. Specifically, from the layout of ERP, we first compute the spherical angles (azimuth and polar) of each pixel in the camera-centric spherical coordinates. After that, we expand this two-channel angle field into the image feature dimension using sine-cosine basis embedding, forming the Spherical Embedding. Since all panoramas have the same full FoV, this spherical embedding can be fixed and reusable. Therefore, to inject spherical awareness, it's only necessary to let the image feature "attend" to the spherical embedding, but not vice versa—the spherical embedding doesn't need to be further refined. Consequently, rather than adding positional embeddings onto the image features before self-attention, as in standard ViTs (Vaswani et al., 2017; Dosovitskiy et al., 2020), SphereViT uses cross-attention: image features are regarded as queries and the spherical embeddings as keys and values. This design lets the image feature explicitly attend to the panorama's spherical geometry, yielding distortion-aware representations and improved performance.

---

[1]Please see *Supp*'s Sec. D for discussions on: metric, scale-invariant (biased), and affine-invariant (relative).

[2]We acknowledge the distinction between distance ($d = \sqrt{x^2 + y^2 + z^2}$) and depth ($d = z$). We focus on scale-invariant distance prediction. Please allow us to use "depth" occasionally for readability and fluency.

[3]ERP can represent a full vertical FoV (*i.e.,* $180°$). If smaller than $180°$, cylindrical projection can be used, such as the panoramic camera mode in mobile phones. Both can present a full horizontal FoV (*i.e.,* $360°$).

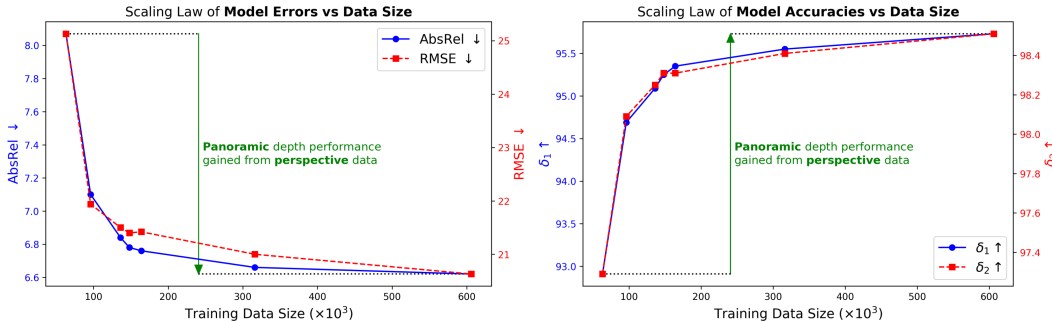

Figure 2: Scaling-law curves of model performance vs data size. Native, high-quality panoramic data is scarce, constraining the zero-shot generalization of panoramic depth estimators. With our data curation engine, DA$^2$ achieves steadily and clearly higher performance as more perspective depth data are converted to panoramic form. Detailed numerical results are provided in Tab. 2.

To validate DA$^2$, we conduct a comprehensive benchmark on scale-invariant distance combining multiple well-recognized evaluation datasets. However, due to the scarcity of panoramic data, existing zero-shot approaches in panoramic depth estimation are limited, whereas in perspective, there exist many powerful zero-shot methods. Therefore, to ensure a more fair and comprehensive comparison, following the panoramic depth estimation pipeline proposed by Wang et al. (2025c;d), we also benchmark DA$^2$ against prior zero-shot perspective depth estimators (Hu et al., 2024; Yin et al., 2023; Piccinelli et al., 2024; 2025b; Wang et al., 2025a;c;d; Bhat et al., 2023; Yang et al., 2024a;b; He et al., 2024b), The results in Tab. 1 clearly demonstrate DA$^2$'s SoTA performance, with an average 38% improvement on AbsRel over the strongest zero-shot baseline. Notably, it even surpasses prior in-domain methods, further underscoring its superior generalization ability. Beyond that, DA$^2$ seamlessly supports various applications, such as panoramic multi-view reconstruction, home decoration, and robotics simulation (please see our *Supp*'s Sec. A). **Our key contributions are:**

- **Panoramic data curation engine.** We introduce a data curation engine that generates high-quality panoramic depth data from perspective data, greatly scaling up the panoramic depth training data and substantially improving the zero-shot generalization ability of DA$^2$.

- **SphereViT.** We propose SphereViT—the primary backbone of DA$^2$. By directly leveraging the spherical coordinates of panoramas, SphereViT effectively mitigates the impact of spherical distortions and enhances the spherical geometry awareness of image features.

- **Comprehensive benchmark.** Both zero-shot / in-domain, panoramic / perspective methods are compared to build a comprehensive benchmark for panoramic depth estimation.

- **SoTA performance.** Experimental results clearly demonstrate DA$^2$'s SoTA performance. DA$^2$ even beats prior in-domain methods. It also enables many downstream applications.

## 2 RELATED WORKS

### 2.1 PERSPECTIVE DEPTH ESTIMATION

Perspective depth estimation is being advanced very rapidly. Metric and scale-invariant depth models, driven by large-scale training data, have achieved strong results, like UniDepth (Piccinelli et al., 2024; 2025b), Metric3D (Hu et al., 2024; Yin et al., 2023), DepthPro (Bochkovskiy et al., 2025), and MoGe (Wang et al., 2025c;d). Relative depth models also benefit greatly from scaling up the training data, like DepthAnything (Yang et al., 2024a;b). Another line of work fine-tunes massively pre-trained generative models, *e.g.*, Stable Diffusion (Rombach et al., 2022; Ho et al., 2020; He et al., 2024a; Li et al., 2024b; Liang et al., 2024; Gu et al., 2024), FLUX (BFL, 2024; Yang et al., 2025b), with limited high-quality data, also yielding impressive results (Ke et al., 2024; He et al., 2024b; Wang et al., 2025b; Li et al., 2024a). Despite these remarkable advances, perspective methods remain constrained by the limited FoV and cannot estimate depth in *all directions* simultaneously. In contrast, DA$^2$ targets full FoV depth estimation with strong zero-shot generalization.

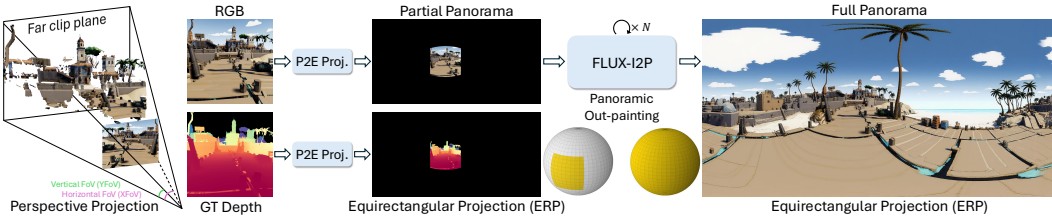

Figure 3: Panoramic data curation engine. This module converts large-scale, high-quality perspective RGB–depth pairs into *full* panoramas through P2E projection and panoramic out-painting using FLUX-I2P. It dramatically scales up the panoramic depth training data, forming a solid training data foundation for DA². The highlighted area on the spheres indicate the FoV coverage.

## 2.2 PANORAMIC DEPTH ESTIMATION

**In-domain.** Due to the scarcity of panoramic data, most existing methods are constrained to in-domain settings. Network designs have evolved from CNNs (Zioulis et al., 2018; Zhuang et al., 2022)) to ViTs (Shen et al., 2022; Yun et al., 2023). Pipeline designs are mainly aimed to mitigate the spherical distortions inherent in panoramas. Many approaches fuse features from both the ERP (1 panorama) and cubemap (6 perspectives) projections (Wang et al., 2020; Jiang et al., 2021; Wang et al., 2022; Li et al., 2022; Ai et al., 2023; Wang & Liu, 2024). For alternative solutions, SliceNet (Pintore et al., 2021) and HoHoNet (Sun et al., 2021) use RNNs or LSTMs along longitudes. SphereDepth (Yan et al., 2022), Elite360D (Ai & Wang, 2024), HUSH (Lee et al., 2025) introduce spherical icosahedral meshes and spherical harmonics. While effective, these strategies still require additional modules, making them less streamlined and efficient. DA² introduces SphereViT to handle the spherical distortions in an end-to-end manner, without extra modules.

**Zero-shot.** With the rise of zero-shot perspective depth estimators, there has been a trend toward developing zero-shot depth estimators for panoramas. 360MonoDepth (Rey et al., 2022) blends tangent perspective depths predicted by MiDaS (Ranftl et al., 2020) on an icosahedral mesh, but suffers from multi-view inconsistencies. PanDA (Cao et al., 2025) leverages Möbius transformation-based data augmentation for self-supervision. UniK3D Piccinelli et al. (2025a) separately predicts camera rays and distance maps, can generalize on various cameras. But their performance remains suboptimal, due to limited panoramic data: ∼20K labeled and ∼92K unlabeled in PanDA, ∼29K in UniK3D. DepthAnyCamera (Guo et al., 2025) projects perspective images with various horizontal FoVs ($20°-124°$, $\ll 360°$) into spherical space, can also generalize on various cameras. But its performance still remains constrained by the incomplete FoVs. In contrast, DA² introduces a panoramic data curation engine, significantly boosting the quantity and diversity of panoramic data from available perspective data, yielding a clearly enhanced zero-shot generalization performance.

## 3 METHODOLOGY

This section presents the methodology of DA² in detail, covering the panoramic data curation engine (Sec. 3.1) and SphereViT with its training loss functions (Sec. 3.2).

### 3.1 PANORAMIC DATA CURATION ENGINE

*"The quality of your data determines the ceiling of your ambitions."* (Surge AI, 2020)

Due to the scarcity of high-quality panoramic data, existing panoramic depth estimators are often trained and evaluated within specific domains, greatly restricting their zero-shot generalization ability and real-world applicability. Thus, the very first goal of this work is to scale up the panoramic data and build a strong data foundation for DA². Motivated by this, we propose a perspective-to-panoramic data curation engine that generates high-quality panoramic data from perspective data.

As illustrated in Fig. 3, the inputs of the panoramic data curation engine are a perspective image sized $(W_{per}, H_{per})$ and its FoVs, *i.e.*, XFoV and YFoV. XFoV represents the coverage of this perspective image in the azimuth field $|\phi_l - \phi_r|$ and YFoV denotes the coverage in the polar angle field $|\theta_u - \theta_d|$. At first, P2E projection will be performed to map the perspective image onto the spherical space.

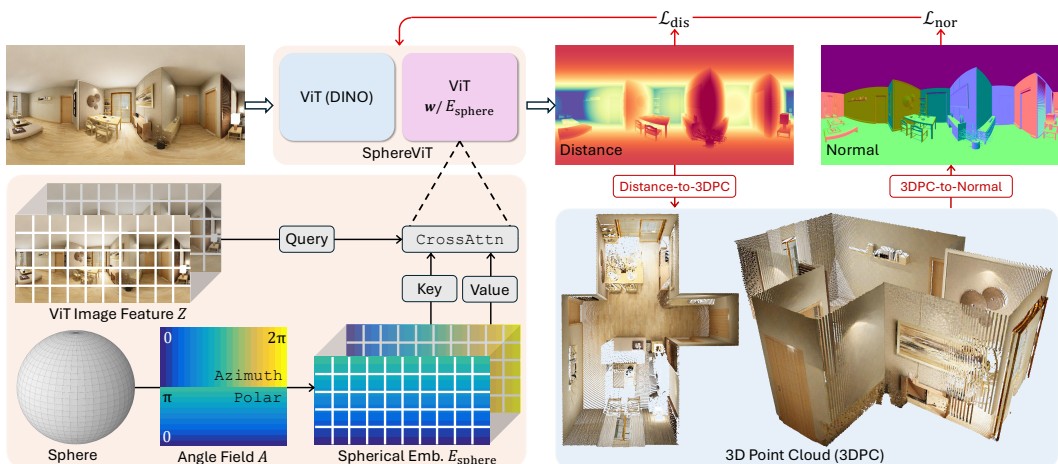

Figure 4: The architecture of SphereViT and training losses. By leveraging the spherical embedding $E_{\text{sphere}}$, which is explicitly derived from the spherical coordinates of panoramas, SphereViT produces distortion-aware image features, yielding more accurate geometrical estimation for panoramas. The training supervision combines a distance loss $\mathcal{L}_{\text{dis}}$ for globally accurate distance values and a normal loss $\mathcal{L}_{\text{nor}}$ for locally smooth and sharp surfaces. The effect of $\mathcal{L}_{\text{nor}}$ is ablated in Fig. 6 (b) and Tab. 3.

Specifically, we start by obtaining the focal lengths from both FoVs:

$$f_x = \frac{W_{\text{per}}}{2 \times \tan\left(\frac{\text{FoV}_x}{2}\right)}, \quad f_y = \frac{H_{\text{per}}}{2 \times \tan\left(\frac{\text{FoV}_y}{2}\right)}. \tag{1}$$

Then, the 3D vector $\mathbf{d}$ and its unit vector $\hat{\mathbf{d}}$ from the perspective camera to each 2D pixel $(x, y)$ of the perspective image ($x \in [0, W_{\text{per}} - 1], y \in [0, H_{\text{per}} - 1]$) are given by:

$$\mathbf{d} = [\frac{(x - \frac{W_{\text{per}}-1}{2})}{f_x}, \frac{(y - \frac{H_{\text{per}}-1}{2})}{f_y}, 1], \quad \hat{\mathbf{d}} = \frac{\mathbf{d}}{|\mathbf{d}|}. \tag{2}$$

Then, in the *spherical space*, the azimuth $\phi$ (longitude) and polar $\theta$ (colatitude) angles of $\hat{\mathbf{d}}$ are:

$$\phi = \text{atan2}(\hat{\mathbf{d}}_x, \hat{\mathbf{d}}_z) + \phi_c, \quad \theta = \arccos(\hat{\mathbf{d}}_y) + \theta_c, \tag{3}$$

where $(\phi_c, \theta_c)$ denote the spherical coordinates of the perspective image's optical center, used as offsets to obtain the absolute longitude and colatitude of each pixel. After that, the mapped pixel position $(u, v)$ on the ERP image (*i.e.*, panorama) sized $(W_{\text{pano}}, H_{\text{pano}})$ is given by:

$$u = \frac{\phi}{2\pi} W_{\text{pano}}, \quad v = \frac{\theta}{\pi} H_{\text{pano}}, \tag{4}$$

where $\phi \in [0, 2\pi), \theta \in [0, \pi]$. After P2E projection, due to the limited FoV of perspective images, only a small portion of the sphere can be covered, as highlighted in Fig. 3's left sphere. This incompleteness leads to suboptimal performance: 1) the model lacks global context since it never observes the full views of panoramic images, particularly near the poles; and 2) spherical distortions vary significantly between the equator and poles, with severe stretching occurring at high latitudes.

Thus, following (Tencent, 2025), the second step of our data curation engine adopts a LoRA (Hu et al., 2022) fine-tuned FLUX model named FLUX-I2P for panoramic out-painting, generating "full" panoramas from the "partial" panoramas. Earlier panoramic out-painting methods (Gao et al., 2024; Feng et al., 2023) often exhibited spatial inconsistencies, especially near the poles and the left–right seam. To address this, FLUX-I2P concatenates image features with the spherical coordinates (azimuth $\phi$ and polar $\theta$) along the channel dimension before feeding them into the Diffusion Transformer (DiT) (Peebles & Xie, 2022), to improve the spatial coherence. For the GT depth associated with the perspective image, we apply only the P2E projection *without* panoramic out-painting, because the *absolute accuracy* of out-painted depth is hard to guarantee. As ablated in Tab. 3, although the panoramic out-painting on the P2E projected GT depth is not performed, FLUX-I2P's panoramic out-painting on the RGB images clearly improves the panoramic depth estimation performance by a large margin, demonstrating its significance in our panoramic data curation engine.

## 3.2 SphereViT & Training Losses

This data curation engine creates $\sim$543K panoramic samples, scales the total from $\sim$63K to $\sim$607K ($\sim$10 times), significantly addressing the data scarcity issue that causes poor generalization. Here we focus on DA$^2$'s model structure and training, to effectively learn from the greatly scaled-up data.

Recently, ViT-based depth models have achieved great success (Wang et al., 2025c;d; Yang et al., 2024a;b; Piccinelli et al., 2025a), where positional embeddings (PE) are crucial for encoding spatial information. For perspectives, PE is typically derived from the 2D $(x, y)$ pixel coordinates. However, for panoramas, pixel coordinates $(u, v)$ correspond to spherical coordinates (longitude $\phi$ and latitude $\theta$). The spherical nature introduces non-uniformity: high-latitude regions (near the poles) are stretched, while low-latitude regions (near the equator) are compressed. Conventional 2D PE cannot account for this spherical distortion, limiting the model's spherical spatial understanding. To address this, many approaches fuse features from both the ERP (1 panorama) and cubemap (6 perspectives) projections or employ auxiliary modules, introducing inefficiencies and complexity. In contrast, DA$^2$ aims to handle the distortions more simply and efficiently, without extra modules.

To this end, DA$^2$ proposes SphereViT, as illustrated in Fig. 4. SphereViT leverages the spherical coordinates of panoramas to efficiently and explicitly inject spherical-awareness into the ViT image features, yielding distortion-aware representations and improved performance. Specifically, we first compute the azimuth and polar angles $(\phi, \theta)$ of each pixel $(u, v)$ in an ERP image sized $(W, H)$:

$$\phi = 2\pi \times \frac{u}{W}, \quad \theta = \pi \times \frac{v}{H}. \tag{5}$$

Then, given the image feature $Z \in \mathbb{R}^{(H' \times W') \times D}$, where $W' = \frac{W}{P}, H' = \frac{H}{P}$ and $P$ is patch size, we resize and flatten this two-channel angle field $A \in \mathbb{R}^{H \times W \times 2}$ (Eq. 5) into $A' \in \mathbb{R}^{(H' \times W') \times 2}$. Motivated by the PE mechanism of ViT, sine-cosine embedding is utilized to expand $A'$'s channel from 2 to the image feature dimension $D$. Concretely, we first define a series of coefficients $\{2^{d_n}\}_{n=1}^{D'}$, where $D' = \frac{D}{4}, d_n = \frac{(n-1)\log_2(H')}{D'}$. Then, for each two-channel unit of $A'$: $A'_{i,j} = [\phi_i, \theta_j] \in \mathbb{R}^{1 \times 2}$, where $i \in [0, W' - 1], j \in [0, H' - 1]$, we transpose and multiple it with the coefficients:

$$\begin{bmatrix} \phi_i \\ \theta_j \end{bmatrix} \times \begin{bmatrix} 2^{d_1} & 2^{d_2} & \cdots & 2^{d_{D'}} \end{bmatrix} = \begin{bmatrix} 2^{d_1}\phi_i & 2^{d_2}\phi_i & \cdots & 2^{d_{D'}}\phi_i \\ 2^{d_1}\theta_j & 2^{d_2}\theta_j & \cdots & 2^{d_{D'}}\theta_j \end{bmatrix}. \tag{6}$$

Eq. 6's result is shaped $2 \times D'$. We then apply the sine-cosine embedding on each unit of this matrix:

$$\begin{bmatrix} [\sin(2^{d_1}\phi_i), \cos(2^{d_1}\phi_i)]^\top & [\sin(2^{d_2}\phi_i), \cos(2^{d_2}\phi_i)]^\top & \cdots & [\sin(2^{d_{D'}}\phi_i), \cos(2^{d_{D'}}\phi_i)]^\top \\ [\sin(2^{d_1}\theta_j), \cos(2^{d_1}\theta_j)]^\top & [\sin(2^{d_2}\theta_j), \cos(2^{d_2}\theta_j)]^\top & \cdots & [\sin(2^{d_{D'}}\theta_j), \cos(2^{d_{D'}}\theta_j)]^\top \end{bmatrix}. \tag{7}$$

Eq. 7 has a shape of $2 \times D' \times 2$. Now, the flattened transformation of Eq. 7—with a dimension of $2 \times D' \times 2 = D$—is the unit $(i, j)$ of the Spherical Embedding $E_{\text{sphere}} \in \mathbb{R}^{(H' \times W') \times D}$.

As discussed in Sec. 1, all panoramas share the same $360° \times 180°$ FoV, so the spherical embedding is fixed, reusable, and doesn't need to be further refined. Thus, to inject spherical awareness, it's only necessary to let image features $Z$ "attend" to the embedding $E_{\text{sphere}}$, but not vice versa. Accordingly, SphereViT replaces the usual self-attention (after addition: $Z + E_{\text{sphere}}$) with cross-attention, where image features $Z$ serve as queries and the spherical embeddings $E_{\text{sphere}}$ act as keys and values:

$$\texttt{CrossAttn}\,(Z, E_{\text{sphere}}) = \text{SoftMax}\left(\frac{ZW_Q\,(E_{\text{sphere}}\,W_K)^\top}{\sqrt{D_k}}\right)(E_{\text{sphere}}\,W_V), \tag{8}$$

where $W_Q, W_K, W_V \in \mathbb{R}^{D \times D_k}$ are learnable projection matrixs, and $Z, E_{\text{sphere}} \in \mathbb{R}^{(H' \times W') \times D}$. This cross-attention with spherical embedding $E_{\text{sphere}}$ allows the image features $Z$ to "learn" the underlying spherical structures of the panoramas, producing distortion-aware representations and leading to clearly enhanced geometrical fidelity as demonstrated in Fig. 6 (a) and Tab. 3.

**Training Losses.** DA$^2$'s SphereViT is trained end-to-end to estimate dense, scale-invariant distance $\hat{D} \in \mathbb{R}^{H \times W}$ from a panoramic RGB input $I \in \mathbb{R}^{H \times W \times 3}$. The supervision combines two terms: a distance loss $\mathcal{L}_{\text{dis}}$ that enforces globally accurate distance values, and a normal loss $\mathcal{L}_{\text{nor}}$ that promotes locally smooth, sharp geometrical surfaces, especially in regions where distance values are similar but surface normals vary significantly. Concretely, let $\hat{D}$ and $D^\star$ be the predicted and GT

Table 1: Quantitative comparison. For a fair and comprehensive benchmark, we include both zero-shot / in-domain, panoramic / perspective approaches. The ==best== and second best performances are highlighted (in *zero-shot* setting). In *all* settings (both zero-shot and in-domain), the **best** and second best performances are bolded and underlined. DA² outperforms all other methods no matter in zero-shot or all settings, particularly showing large gains under the zero-shot setting. Median alignment (scale-invariant) is adopted by default. △: Affine-invariant alignment (scale and shift-invariant), for prior *relative* depth estimators: DepthAnything v1v2 (Yang et al., 2024a;b), Lotus (He et al., 2024b), and PanDA (Cao et al., 2025). We also report PanDA's results in median alignment for fairness. ⋆: Implemented by ourselves (code will be released). The unit is percentage (%).

| Categories | Method | Stanford2D3D | | | | Matterport3D | | | | PanoSUNCG | | | | Rank↓ Zero-shot | Rank↓ All |
|---|---|---|---|---|---|---|---|---|---|---|---|---|---|---|---|
| | | AbsRel↓ | RMSE↓ | $\delta_1 \uparrow$ | $\delta_2 \uparrow$ | AbsRel↓ | RMSE↓ | $\delta_1 \uparrow$ | $\delta_2 \uparrow$ | AbsRel↓ | RMSE↓ | $\delta_1 \uparrow$ | $\delta_2 \uparrow$ | | |
| In-domain | OmniDepth | 19.96 | 61.52 | 68.77 | 88.91 | 29.01 | 76.43 | 68.30 | 87.94 | 11.43 | 37.10 | 87.05 | 93.65 | – | 26.33 |
| | FCRN | 18.37 | 57.74 | 72.30 | 92.07 | 24.09 | 67.04 | 77.03 | 91.74 | 9.79 | 39.73 | 92.23 | 96.59 | – | 24.00 |
| | BiFuse | 12.09 | 41.42 | 86.60 | 95.80 | 20.48 | 62.59 | 84.52 | 93.19 | 5.92 | 25.96 | 95.90 | 98.23 | – | 16.83 |
| | EGFormer | 15.28 | 49.74 | 81.85 | 93.38 | 14.73 | 60.25 | 81.58 | 93.90 | – | – | – | – | – | 15.50 |
| | SliceNet | 12.49 | 43.70 | 83.77 | 94.14 | 17.64 | 61.33 | 87.16 | 94.83 | – | – | – | – | – | 14.17 |
| | SphereDepth | 11.58 | 45.12 | 86.66 | 96.42 | 12.05 | 59.22 | 86.20 | 95.19 | – | – | – | – | – | 12.42 |
| | BiFuse++ | 11.17 | 37.20 | 87.83 | 96.49 | 14.24 | 51.90 | 87.90 | 95.17 | **5.24** | 24.77 | **96.30** | 98.35 | – | 12.17 |
| | UniFuse | 11.14 | 36.91 | 87.11 | 96.64 | 10.63 | 49.41 | 88.97 | 96.23 | _5.28_ | 27.04 | 95.91 | 98.25 | – | 11.25 |
| | HoHoNet | 10.14 | 38.34 | 90.54 | 96.93 | 14.88 | 51.38 | 87.86 | 95.19 | – | – | – | – | – | 10.33 |
| | Elite360D | 11.82 | 37.56 | 88.72 | 96.84 | 11.15 | 48.75 | 88.15 | 96.46 | – | – | – | – | – | 10.00 |
| | PanoFormer | 11.31 | 35.57 | 88.08 | 96.23 | 9.04 | 44.70 | 88.16 | 96.61 | 5.34 | **18.90** | 94.87 | **98.83** | – | 9.50 |
| | HRDFuse | 9.35 | 31.06 | 91.40 | 97.98 | 9.67 | 44.33 | 91.62 | 96.69 | 6.90 | 27.44 | 92.15 | 97.42 | – | 9.50 |
| | SphereFusion | 8.99 | 31.94 | 92.57 | 97.55 | 11.45 | 48.85 | 87.01 | 96.13 | – | – | – | – | – | 7.92 |
| | ACDNet | 9.84 | 34.10 | 88.72 | 97.04 | 10.10 | 46.29 | 90.00 | 96.78 | – | – | – | – | – | 7.08 |
| | DepthAnywhere | 11.80 | 35.10 | 91.00 | 97.10 | 8.50 | – | 91.70 | 97.60 | – | – | – | – | – | 5.33 |
| | OmniFusion | 9.50 | 34.74 | 89.88 | 97.69 | 9.00 | 42.61 | 91.89 | 97.97 | – | – | – | – | – | 5.00 |
| | HUSH | _7.82_ | 33.32 | _93.84_ | **98.49** | _8.38_ | 41.64 | 92.87 | 96.98 | – | – | – | – | – | _3.67_ |
| Zero-shot (fusion) | Lotus-D⋆△ | 45.88 | 48.86 | 37.67 | 68.39 | 32.39 | 85.86 | 48.15 | 78.23 | 37.96 | 77.02 | 46.08 | 77.41 | 17.00 | 30.33 |
| | Lotus-G⋆△ | 45.08 | 47.90 | 38.38 | 69.18 | 31.82 | 84.51 | 49.11 | 78.92 | 38.02 | 76.82 | 46.16 | 77.51 | 16.17 | 29.50 |
| | DepthAnything⋆△ | 37.21 | 43.41 | 47.08 | 76.93 | 24.46 | 60.54 | 58.32 | 87.09 | 24.58 | 52.22 | 64.86 | 90.39 | 14.58 | 27.42 |
| | DepthAnythingv2⋆△ | 36.79 | 43.39 | 47.66 | 76.96 | 25.85 | 70.67 | 58.42 | 86.19 | 23.90 | 50.74 | 66.86 | 90.89 | 14.25 | 27.25 |
| | ZoeDepth⋆ | 17.60 | 33.74 | 74.26 | 92.86 | 18.43 | 53.46 | 72.18 | 93.12 | 21.16 | 44.81 | 69.34 | 94.45 | 11.75 | 22.58 |
| | 360MonoDepth | 16.50 | 28.23 | 74.56 | 92.98 | 20.83 | 79.09 | 65.58 | 88.95 | 11.43 | 28.29 | 90.75 | 98.12 | 10.83 | 21.67 |
| | VGGT⋆ | 18.70 | 33.50 | 74.08 | 83.90 | 10.78 | 30.80 | 88.07 | 97.72 | 8.43 | 25.67 | 94.04 | 98.19 | 8.42 | 15.08 |
| | Metric3D⋆ | 12.93 | 20.80 | 84.77 | 96.52 | 14.11 | 45.11 | 83.09 | 96.59 | 11.42 | 26.95 | 90.45 | 97.33 | 7.67 | 15.17 |
| | UniDepth⋆ | 15.06 | 20.48 | 76.99 | 90.34 | 11.12 | 36.20 | 88.66 | 97.94 | 10.40 | 27.29 | 92.59 | 98.00 | 7.50 | 13.92 |
| | MoGe | 15.81 | 25.76 | 79.02 | 83.32 | 10.04 | 35.91 | 90.80 | 98.45 | 8.60 | 25.80 | 93.85 | 98.31 | 6.33 | 12.08 |
| | UniDepthv2⋆ | 13.08 | 20.46 | 82.12 | 89.21 | 10.86 | 37.68 | 88.76 | 97.86 | 9.74 | 25.94 | 93.06 | 98.30 | 6.25 | 12.17 |
| | Metric3Dv2⋆ | 11.59 | 21.78 | 86.07 | 97.36 | 17.78 | 60.20 | 75.35 | 93.22 | 7.30 | 24.54 | 94.25 | 98.25 | 6.08 | 14.08 |
| | MoGev2 | 14.69 | 24.24 | 79.98 | 84.39 | 10.34 | 36.91 | 89.48 | 98.24 | 8.26 | 24.67 | 94.15 | 98.52 | 5.58 | 11.25 |
| Zero-shot (end2end) | PanDA | 48.44 | 53.06 | 33.92 | 51.33 | 37.10 | 101.5 | 42.51 | 67.29 | 34.73 | 79.69 | 44.49 | 71.45 | 17.50 | 30.83 |
| | DepthAnyCamera | 15.26 | 22.80 | 75.47 | 92.90 | 15.60 | 61.85 | 77.27 | 95.62 | 12.78 | 27.88 | 89.67 | 97.85 | 9.75 | 19.42 |
| | PanDA△ | 16.48 | 23.64 | 73.26 | 85.42 | 8.88 | 33.25 | 92.09 | 98.26 | 6.71 | 21.85 | 95.42 | 98.25 | 5.33 | 10.33 |
| | UniK3D | 11.31 | _19.72_ | 88.94 | 95.33 | 9.66 | _32.66_ | _93.00_ | _98.58_ | 11.46 | 25.38 | 90.18 | 98.02 | 4.58 | 8.75 |
| | **DA² (Ours)** | **7.23** | **14.00** | **95.45** | _98.38_ | **6.67** | **28.82** | **95.61** | **98.60** | 5.96 | _19.07_ | 96.12 | _98.55_ | 1.00 | **1.67** |

distances. Then the surface normals can be obtained with a distance-to-normal operator $\texttt{D2N}$, giving $\hat{N} = \texttt{D2N}(\hat{D})$ and $N^\star = \texttt{D2N}(D^\star)$ when GT normals are not directly available. Since we focuses on scale-invariant distance, $\hat{D}$ is median-aligned before loss computing: $\hat{D}^{\text{med}} = \hat{D} \times \frac{\text{Median}(D^\star)}{\text{Median}(\hat{D})}$. While training the SphereViT, we minimize the per-pixel L1 difference for both $\mathcal{L}_{\text{dis}}$ and $\mathcal{L}_{\text{nor}}$:

$$\mathcal{L}_{\text{dis}} = \frac{1}{|\Omega|} \sum_{p \in \Omega} \big| \hat{D}_p^{\text{med}} - D_p^\star \big|, \quad \mathcal{L}_{\text{nor}} = \frac{1}{|\Omega|} \sum_{p \in \Omega} \big| \hat{N}_p - N_p^\star \big|, \tag{9}$$

where $\Omega$ is the set of valid pixels. For $\mathcal{L}_{\text{nor}}$, we prefer the L1 norm over the commonly-used angular discrepancy $1 - \langle \hat{N}_p, N_p \rangle$ as the latter may introduce gradient collapse and destabilize training. The total loss is a weighted sum: $\mathcal{L} = \lambda_{\text{d}}\mathcal{L}_{\text{dis}} + \lambda_{\text{n}}\mathcal{L}_{\text{nor}}$, where $\lambda_{\text{d}}$ and $\lambda_{\text{n}}$ are scalar weights.

# 4 EXPERIMENTS

## 4.1 EXPERIMENTAL SETTINGS

**Training Datasets.** DA² is trained using 7 high-quality datasets. 6 perspective: Hypersim (Roberts et al., 2021), Virtual-KITTI 2 (Cabon et al., 2020), MVS-Synth (Huang et al., 2018), Unreal-

Table 2: Ablation study on training data scaling. The results show clear, steady performance gains as the size of training data grows. $^{\text{Pano}}$ indicates a perspective dataset converted into panoramic through our data curation engine. The average results across multiple datasets are reported (also in Tab. 3). Please see *Supp*'s Sec. B for more discussions about the data curation engine and the curated data.

| S3D | HPS$^{\text{Pano}}$ | VK$^{\text{Pano}}$ | MVS$^{\text{Pano}}$ | US4K$^{\text{Pano}}$ | 3DKB$^{\text{Pano}}$ | DR$^{\text{Pano}}$ | Data Size | AbsRel↓ | RMSE↓ | $\delta_1$ ↑ | $\delta_2$ ↑ |
|---|---|---|---|---|---|---|---|---|---|---|---|
| ✓ | ✗ | ✗ | ✗ | ✗ | ✗ | ✗ | 63,097 | 8.07 | 25.13 | 92.91 | 97.29 |
| ✓ | ✓ | ✗ | ✗ | ✗ | ✗ | ✗ | 96,677 | 7.10 | 21.94 | 94.69 | 98.09 |
| ✓ | ✓ | ✓ | ✗ | ✗ | ✗ | ✗ | 136,326 | 6.84 | 21.50 | 95.09 | 98.25 |
| ✓ | ✓ | ✓ | ✓ | ✗ | ✗ | ✗ | 148,326 | 6.78 | 21.40 | 95.25 | 98.31 |
| ✓ | ✓ | ✓ | ✓ | ✓ | ✗ | ✗ | 164,726 | 6.76 | 21.42 | 95.35 | 98.31 |
| ✓ | ✓ | ✓ | ✓ | ✓ | ✓ | ✗ | 316,722 | 6.66 | 21.00 | 95.55 | 98.41 |
| ✓ | ✓ | ✓ | ✓ | ✓ | ✓ | ✓ | 606,522 | **6.62** | **20.63** | **95.73** | **98.51** |

Stereo4K (Tosi et al., 2021), 3D-Ken-Burns (Niklaus et al., 2019), Dynamic Replica (Karaev et al., 2023), totaling 543,425 samples; 1 panoramic: Structured3D (Zheng et al., 2020) (63,097 samples).

**Evaluation Datasets & Metrics.** For a fair and reproducible comparison, DA$^2$ is evaluated on three widely-used, well-recognized benchmarks in panoramic depth estimation: Stanford2D3D-S (Armeni et al., 2017) (all splits), Matterport3D (Chang et al., 2017) (test split), and PanoSUNCG (Wang et al., 2018) (test split), using 2 error metrics (AbsRel, RMSE), and 2 accuracy metrics ($\delta_1$, $\delta_2$). Please see the implementation details (Sec. E) and metric formulations (Sec. C) in our *Supp*.

## 4.2 QUANTITATIVE & QUALITATIVE COMPARISONS

Tab. 1 presents a comprehensive comparison of DA$^2$ with previous SoTA approaches. Following (Wang et al., 2025c;d), we also include prior perspective methods for a more thorough comparison. As demonstrated in Tab. 1, DA$^2$ consistently outperforms all other methods across various settings. Particularly in the zero-shot setting, DA$^2$ shows significant gains over the second-best method by an average of 38% in AbsRel and 22% in RMSE, achieving a remarkable average $\delta_1$ of 95.73% and $\delta_2$ of 98.51%. Notably, even as a zero-shot model, DA$^2$ surpasses earlier in-domain methods as well, further underscoring its superior zero-shot generalization ability.

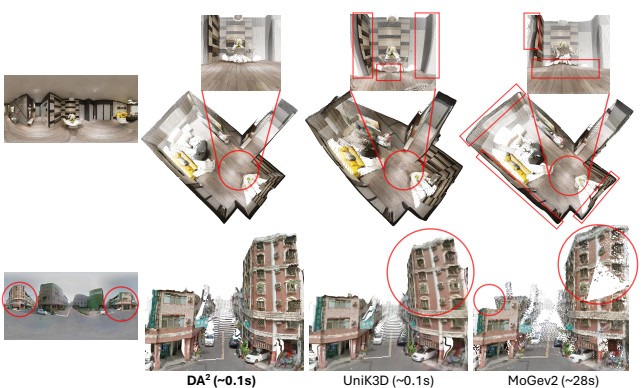

Figure 5: Qualitative comparisons. Compared with UniK3D and MoGev2, DA$^2$ delivers more accurate geometric predictions and, as an end-to-end approach, achieves significantly higher inference efficiency than fusion-based methods.

In addition, for better access the DA$^2$'s performance, we also conduct qualitative comparisons with UniK3D (Piccinelli et al., 2025a)—the strongest prior zero-shot, end-to-end method, and Mo-Gev2 (Wang et al., 2025d)—the strongest prior zero-shot, fusion-based method, as highlighted in Fig. 5. Thanks to our data curation engine, DA$^2$ is trained with about 21× more panoramic data than UniK3D, exhibiting clearly more accurate geometrical predictions. DA$^2$ continuously yields better results over MoGev2, as its panoramic performance is restricted by the multi-view inconsistencies during fusion, *e.g.*, irregular walls, fragmented buildings, etc. We also report the inference times: as an end-to-end method, DA$^2$ achieves significantly higher efficiency than fusion-based approaches.

## 4.3 ABLATION STUDIES

**Training Data.** As reported in Tab. 2, DA$^2$'s performance steadily improves as more perspective depth data converted into panoramic, thanks to our data curation engine. Fig. 2 further shows rapid gains once the curated perspective data is introduced, with performance gradually converging as the data scales. Even near convergence, further improvements are still anticipated with additional data.

Table 3: Ablation studies on: 1) the panoramic out-painting in the data curation engine, 2) spherical embedding $E_{\text{sphere}}$ in the SphereViT, and 3) the auxiliary normal loss $\mathcal{L}_{\text{nor}}$. The results below demonstrate that each design plays a vital role in achieving the final remarkable performance of $DA^2$.

| Pano. Out-painting | Spherical Emb. $E_{\text{sphere}}$ | Normal Loss $\mathcal{L}_{\text{nor}}$ | Data Size | AbsRel↓ | RMSE↓ | $\delta_1$ ↑ | $\delta_2$ ↑ |
|:---:|:---:|:---:|:---:|:---:|:---:|:---:|:---:|
| ✗ | ✓ | ✓ | 606,522 | 7.59 | 23.80 | 94.12 | 97.86 |
| ✓ | ✗ | ✓ | 606,522 | 6.84 | 20.87 | 95.26 | 98.43 |
| ✓ | ✓ | ✗ | 606,522 | 6.99 | 21.53 | 95.25 | 98.37 |
| ✓ | ✓ | ✓ | 606,522 | **6.62** | **20.63** | **95.73** | **98.51** |

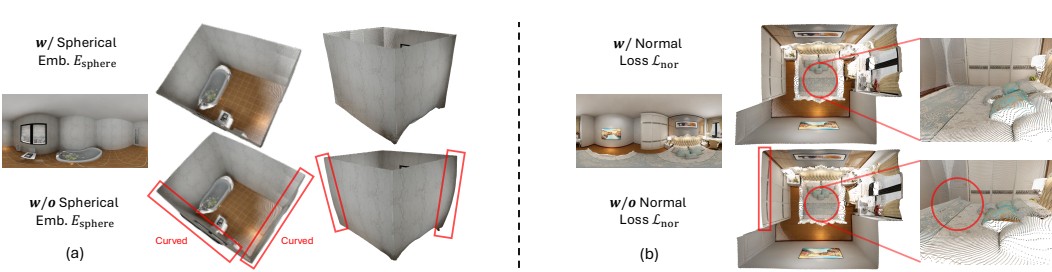

Figure 6: Ablation studies of $DA^2$. (a) Removing the spherical embedding $E_{\text{sphere}}$ causes curved, distorted geometry. (b) Omitting the normal loss $\mathcal{L}_{\text{nor}}$ yields rougher surfaces and more artifacts.

**Panoramic Out-painting.** It is a crucial step in the panoramic data curation engine, generating *full* RGB panoramas from P2E-projected perspective images (Fig. 3). Comparing Tab. 2's 1st row with Tab. 3's 1st row, $DA^2$'s performance can be improved only modestly via scaling up the perspective *w/o* panoramic out-painting, yielding a 0.48 gain in AbsRel. In contrast, incorporating (*w/*) out-painting yields a much larger boost than "*w/o* out-painting" (~3 times), with a 1.45 gain in AbsRel (Tab. 2's 1st row vs. Tab. 3's last row), clearly showing the importance of panoramic out-painting.

**Spherical Embedding.** We here ablate the impact of spherical embedding $E_{\text{sphere}}$ in the SphereViT. As shown in Tab. 3 (2nd vs. last row), including $E_{\text{sphere}}$ noticeably boosts $DA^2$'s performance. Fig. 6 (a) further illustrates that incorporating the spherical embedding produces more accurate geometric understandings on panoramas, while its absence often leads to suboptimal performance (*e.g.*, curved walls), highlighting its effectiveness in mitigating the spherical distortions.

**Training Losses.** We further ablate the auxiliary normal loss $\mathcal{L}_{\text{nor}}$ used for training the SphereViT. As shown in Tab. 3 (3rd vs. last row), adding $\mathcal{L}_{\text{nor}}$ boosts $DA^2$'s performance clearly. Also, as highlighted in Fig. 6 (b), normal supervision yields flatter, smoother, and more coherent geometry, reducing the artifacts that typically appear in ambiguous regions (*e.g.*, corners, edges, and the upper or lower poles), where distance values may be similar but surface normals differ substantially.

# 5 LIMITATION & CONCLUSION

**Limitation.** Despite the strong performance enabled by the large-scale training data thanks to our panoramic data curation engine and distortion-aware SphereViT, $DA^2$ still faces several constraints. As the training resolution (1024×512) is lower than higher-definition formats such as 2K or 4K, and the curated perspective data provide only partially available GT depth in the spherical space, $DA^2$ may occasionally miss fine details (Fig. 7 (a)) and produce visible seams along the panorama's left–right boundaries (which should ideally be seamlessly aligned), as illustrated in Fig. 7 (b).

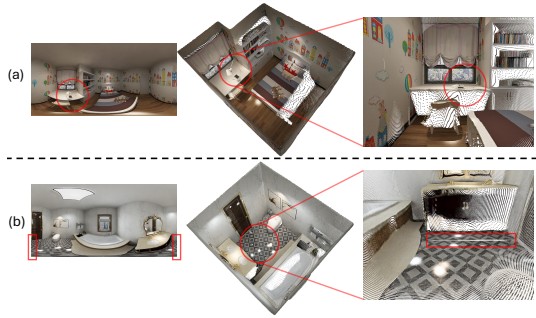

Figure 7: $DA^2$'s limitations. (a) The white lamp's predicted distance is mistakenly aligned with the desk surface. (b) Visible seams appear along the predictions at lower left–right boundaries.

**Conclusion.** We introduce $DA^2$, an end-to-end, zero-shot generalizable, panoramic distance (scale-invariant) estimator that unites a panoramic data curation engine with the distortion-aware Sphere-ViT. Trained on over 600K samples ($\sim$543K curated from perspective and $\sim$63K native panoramas), $DA^2$ delivers SoTA zero-shot performance, outperforming prior methods (both zero-shot and in-domain) by a clear margin while remaining efficient and fully end-to-end. This work shows that scaling up panoramic data and explicitly modeling the spherical geometry enables high-quality and robust $360° \times 180°$ geometrical estimation, paving the way for high-fidelity 3D scene applications, *e.g.*, immersive 3D scene creation, AR/VR, robotics simulation, physical simulation, etc.

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

# SUPPLEMENTARY MATERIALS OF DA$^2$: DEPTH ANYTHING IN ANY DIRECTION

## A APPLICATIONS OF DA$^2$

Leveraging its remarkable capability in zero-shot generalizable panoramic depth estimation, DA$^2$ effectively enables a wide range of 3D reconstruction-related applications.

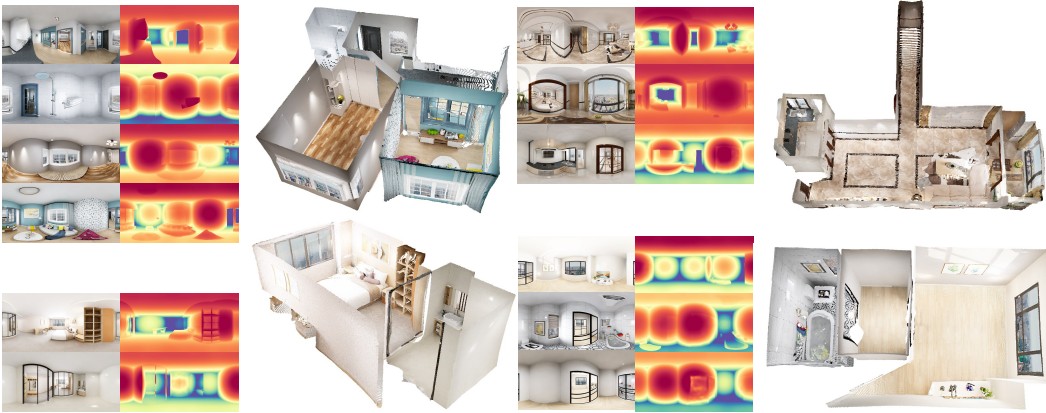

Figure 8: Pano3R: Panoramic Multi-view Reconstruction. Given panoramic images of different rooms from a house / apartment, DA$^2$ enables the reconstruction of a globally aligned 3D point cloud, ensuring the spatial coherence across multiple panoramic views of different rooms.

### A.1 PANO3R: PANORAMIC MULTI-VIEW RECONSTRUCTION

A house / apartment typically consists of multiple distinct rooms, which may exhibit substantial geometric variations. Thanks to the strong zero-shot generalization and high geometric consistency in panoramic depth estimation, DA$^2$ is able to reconstruct a holistic 3D point cloud representation of the indoor layout, leveraging multiple panoramic images captured from different rooms. As shown in Fig. 8, the rooms can be consistently aligned via simple translation, without requiring any scaling or rotation operations. This characteristic highlights the robustness and superior geometric consistency of DA$^2$'s depth estimation, enabling seamless alignment of shared structures such as walls and doors, facilitating applications such as VR-based indoor apartment tours and layout visualization.

**Why is rotation not needed in our cases?**

Of course, thanks to the robustness and high-quality of our panoramic depth estimator, scaling is not needed for alignment. For example, the height of the reconstructed rooms are nearly the same. And in our cases, the rotations are not needed because:

1. The Z-directions of these 360° cameras are vertical to the ground.
2. The directions from the cameras to the central pixel of the multi view panoramas are facing the same direction in the actual 3D space where the multi view panoramas are captured. For example, panorama A and B have a wall and a door overlapped, if the door is located at the left behind A's camera, then it will also be at the right front of B's camera.

### A.2 LAYERED HOME RENOVATION

As illustrated in Fig. 9 (a), given indoor panoramas with three distinct complexity levels—"empty", "simple", and "full"—the multiple sets of 3D point clouds reconstructed from DA$^2$'s panoramic distance maps exhibit high consistency. They can be seamlessly aligned with fine details. As demonstrated in the zoom-in regions of Fig. 9 (a), the fused point clouds are free of distortions: the text on the blackboard is sharp, and the wall boundaries are consistently aligned.

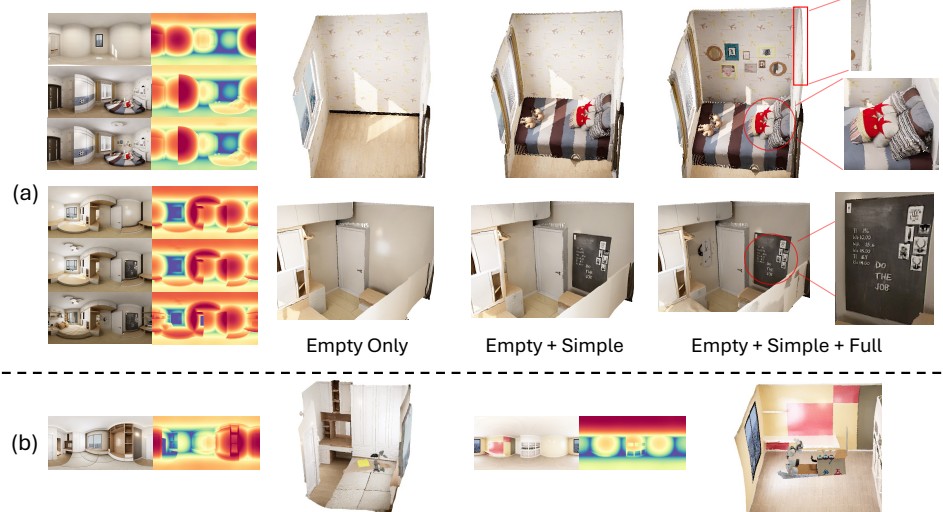

Figure 9: More applications of DA$^2$: (a) Layered Home Renovation. The three input panoramas correspond to different levels of foreground object complexity, denoted as "empty", "simple", and "full". The zoom-in views show that the reconstructed 3D point clouds from these panoramas remain consistently aligned (primarily in backgrounds). (b) Robotics Simulation. The reconstructed 3D point cloud can serve as a practical 3D platform for evaluating robot manipulation performance.

### A.3 ROBOTICS SIMULATION

Benefiting from DA$^2$'s robust panoramic distance estimation, the reconstructed 3D point cloud can serve as a reliable 3D simulation environment for robot manipulation. As illustrated in Fig. 9 (b), it provides a practical 3D platform for simulating and demonstrating robotic tasks.

## B PANORAMIC DATA CURATION ENGINE (MORE DETAILS)

As discussed in Sec. 4.1, 6 perspective datasets (Hypersim (Roberts et al., 2021), Virtual-KITTI 2 (Cabon et al., 2020), MVS-Synth Huang et al. (2018), UnrealStereo4K (Tosi et al., 2021), 3D-Ken-Burns (Niklaus et al., 2019), Dynamic Replica (Karaev et al., 2023)) are transformed into panoramic via the proposed panoramic data curation engine. The curated datasets are summarized in Tab. 4. As shown, the sampling probabilities are normalized across datasets primarily considering data size to ensure a balanced influence during DA$^2$'s training process. Each dataset represents a domain, this balanced mixture ensures DA$^2$'s performance will not be over influenced by a few strong datasets, achieving stable scaling behavior across datasets, and optimal cross-domain generalization. To this end, our data curation engine generates ∼543K high-quality panoramic image–depth pairs from perspective data, expanding the total dataset to ∼607K samples. This substantially enriches the quantity and diversity of panoramic data, constructs a solid data foundation for DA$^2$, and in turn significantly enhances the zero-shot performance of DA$^2$, as demonstrated in Fig. 2 and Tab. 2.

On average, most perspective data samples we used have an X FoV of $70° \sim 90°$ and the height usually equals the width. Thus, the fraction of GT depth labels in the curated panoramas is: $80/180 * 80/360 \approx 9.9\%$. By latitude, the GT depth labels range from $-40°$ to $40°$. Moreover, though the fraction in area is only 10%, the fraction in semantic is much larger, because the ERP images contain much richer semantics near the equator than the poles.

Table 4: Perspective datasets processed by the panoramic data curation engine. For each dataset, the vertical FoV (YFoV) is derived directly from the horizontal FoV (XFoV) as YFoV = XFoV $\times \frac{H}{W}$, where $(W, H)$ denotes the input panorama's width and height.

| Category | Dataset Name | Abbreviation (Tab. 2) | Data Size | In-or-outdoor | XFoV | Sam. Probability |
|---|---|---|---|---|---|---|
| Perspective | Hypersim | HPS | 39,649 | In | $60°$ | 16.59% |
| Perspective | Virtual-KITTI 2 | VK | 33,580 | Out | $80°$ | 14.05% |
| Perspective | MVS-Synth | MVS | 12,000 | Out | $80°$ | 5.02% |
| Perspective | UnrealStereo4K | US4K | 16,400 | Various | $90°$ | 6.86% |
| Perspective | 3D-Ken-Burns | 3DKB | 151,996 | Various | $60°-90°$ | 15.91% |
| Perspective | Dynamic Replica | DR | 289,800 | In | $85°$ | 15.16% |
| Panoramic | Structured3D | S3D | 63,097 | In | $360°$ | 26.41% |

## C EVALUATION METRICS

Concretely, given the predicted panoramic depth $\hat{D}$ and GT $D^\star$, median alignment is performed on the predicted distance $\hat{D}$ before computing the metrics:

$$\hat{D}^{\mathrm{med}} = \hat{D} \times \frac{\mathrm{Median}(D^\star)}{\mathrm{Median}(\hat{D})}, \tag{10}$$

following the evaluation protocols in prior works (Lee et al., 2025; Wang & Liu, 2024; Yun et al., 2023; Yan et al., 2025; 2022; Li et al., 2022; Shen et al., 2022; Zhuang et al., 2022; Wang et al., 2022; Sun et al., 2021; Pintore et al., 2021; Jiang et al., 2021; Wang et al., 2020; Rey et al., 2022; Piccinelli et al., 2025a; Cao et al., 2025), After that, the AbsRel and RMSE are given by:

$$\mathrm{AbsRel} = \frac{1}{|\Omega|} \sum_{p \in \Omega} \frac{|\hat{D}_p^{\mathrm{med}} - D_p^\star|}{D_p^\star}, \quad \mathrm{RMSE} = \frac{1}{|\Omega|} \sqrt{\sum_{p \in \Omega} (\hat{D}_p^{\mathrm{med}} - D_p^\star)^2}, \tag{11}$$

where $\Omega$ is the set of valid pixels. $\delta 1$ and $\delta 2$ denotes the proportion of pixels satisfying $\mathrm{Max}(D_p^\star/\hat{D}_p^{\mathrm{med}}, \hat{D}_p^{\mathrm{med}}/D_p^\star) < 1.25$ and $< 1.25^2$ respectively.

## D DIFFERENCE AMONG: METRIC & SCALE-INVARIANT (BIASED) & AFFINE-INVARIANT (RELATIVE)

**Metric and Scale-invariant Depth.** In depth (or distance) estimation, metric depth $D_{\mathrm{metric}}$ is the strictest setting, where the predicted values correspond to absolute physical distances and can be directly used to reconstruct a "real-scale" point cloud. Scale-invariant (or biased) depth $D_{\mathrm{biased}}$ is still strict but slightly more relaxed than metric: predictions include a global bias or shift, but not in the absolute global scale. Although the depths are not metric, the underlying 3D structure is preserved perfectly (Tab. 5), because the global bias or shift is preserved. During training & evaluation, for scale-invariant depth, median alignment (scale-invariant) is typically adopted to re-scale the underlying 3D structure to real-world size (please see Sec. C). For metric depth, no alignment should ideally be required, but median alignment is still commonly applied because absolute scales can be ambiguous (cameras with different focal lengths can capture visually similar pictures but with substantially different absolute depths) (Hu et al., 2024; Yin et al., 2023; Piccinelli et al., 2025a).

$DA^2$ focuses on panoramic scale-invariant (or biased) distance estimation for two reasons: 1) like metric distance, scale-invariant distance also preserves the full underlying 3D geometry, and 2) $DA^2$ targets on the strong zero-shot generalization across diverse domains, enforcing absolute scales would introduce significant optimization challenges, as indoor and outdoor scenes differ drastically in scale, making the additional cost outweigh the benefits.

**Affine-invariant Depth.** Affine-invariant (or relative) depth $D_{\mathrm{relative}}$ is the loosest definition, much more relaxed than either biased or metric depth, preserving only the "ordering" of depths (which point is closer or farther). Since neither scale nor shift is preserved, affine-invariant depth $D_{\mathrm{relative}}$ cannot be used to reconstruct a *reasonable* 3D point cloud (Tab. 5), but it's useful for tasks where

only relative geometry matters. Affine-invariant alignment (scale and shift-invariant) is usually adopted during training & evaluation of affine-invariant depth estimators. Concretely, given the predicted $\hat{D}_{\text{relative}}$ and GT depth $D^\star$, least squares fitting is performed:

$$\min_{\text{scale, shift}} \sum_{p \in \Omega} \big\| \text{ scale} \times (\hat{D}_{\text{relative},p} + \text{shift}) - D_p^\star \big\|_2^2, \tag{12}$$

where $\Omega$ is the set of valid pixels and the aligned predicted depth is: $\hat{D}^{\text{aff}} = \text{scale} \times (\hat{D}_{\text{relative}} + \text{shift})$.

The summarized difference is listed in Tab. 5. Note that for the "Illustration with $D_{\text{metric}}$" of scale-invariant and affine-invariant depth, we only list the most widely adopted formats, passing over other scales for $D_{\text{biased}}$ and other specific transformations for $D_{\text{relative}}$ like $\exp(\cdot)$ and $\log(\cdot)$.

Table 5: Summarized difference on depth maps among metric, scale-invariant (biased), and affine-invariant (relative). Both metric and scale-invariant depth fully preserve the 3D geometry. Due to the absence of bias or shift, affine-invariant depth is unable to reconstruct an accurate 3D structure.

| Depth Category | Metric Depth | Scale-invariant Depth | Affine-invariant Depth |
|---|---|---|---|
| Illustration with $D_{\text{metric}}$ | $D_{\text{metric}}$ | $\dfrac{D_{\text{metric}}}{\max(D_{\text{metric}})}$ | $\dfrac{D_{\text{metric}} - \min(D_{\text{metric}})}{\max(D_{\text{metric}}) - \min(D_{\text{metric}})}$ |
| 3D Point Cloud on: |  |  |  |

# E   IMPLEMENTATION DETAILS

DA$^2$ is implemented in PyTorch (Paszke et al., 2019). In SphereViT, the backbone of "ViT (DINO)" is initialized from DINOv2-ViT-L (Oquab et al., 2023) with 24 self-attention blocks, following (He et al., 2024b; Ke et al., 2024), to leverage the pre-trained visual priors. The "ViT w/ $E_{\text{sphere}}$" is a lightweight ViT contains only 4 cross-attention blocks. Training the SphereViT takes $\sim$5,000 optimization iterations on 32 NVIDIA H20 GPUs, with a batch size of 768. The distributed training is implemented with Accelerate (Gugger et al., 2022). We set $\lambda_{\text{dis}} = 1.0, \lambda_{\text{nor}} = 2.0$ for balanced loss values. Panoramas and GT depth maps are fed to SphereViT at a resolution of 1024×512. Please see the sampling probabilities of different data sources in Tab. 4. In the panoramic data curation engine, the FLUX-I2P is fine-tuned on FLUX.1 [dev] (BFL, 2024), largely following Tencent (2025). The LoRA rank is set to 256 during the LoRA (Hu et al., 2022) fine-tuning. The positive prompt is: `a clean, realistic, high-quality, high-resolution, panoramic image of a [*] scene`, where `[*]` is either `indoor` or `outdoor`. The negative prompt is: `messy, low-quality, blur, noise, low-resolution, abnormal`. The $\phi_c, \theta_c$ are randomly selected from $\pm 30°$ and $\pm 15°$, respectively. The panoramic out-painting of 543,425 perspective RGB images from various datasets is performed on 64 NVIDIA H20 GPUs and over nearly 9 days. The running time reported in Fig. 5 is tested on a NVIDIA H20 GPU at a resolution of 1024×512, excluding I/O operations.

# F   PRIOR SOTA METHODS FOR COMPARISONS

**In-domain Baselines.** 17 previous in-domain, panoramic depth estimation approaches are selected for the quantitative comparison in Tab. 1: HUSH (Lee et al., 2025), DepthAnywhere (Wang & Liu, 2024), Elite360D (Ai & Wang, 2024), EGFormer (Yun et al., 2023), SphereFusion (Yan et al., 2025), SphereDepth (Yan et al., 2022), OmniFusion (Li et al., 2022), HRDFuse (Ai et al., 2023), PanoFormer (Shen et al., 2022), ACDNet (Zhuang et al., 2022), BiFuse++ (Wang et al., 2022), HoHoNet (Sun et al., 2021), SliceNet (Pintore et al., 2021), UniFuse (Jiang et al., 2021), BiFuse (Wang et al., 2020), FCRN (Laina et al., 2016), and OmniDepth (Zioulis et al., 2018).

**Zero-shot, fusion-based baselines.** 13 zero-shot, fusion-based panoramic depth estimators are selected or implemented. 1 is originally panoramic: 360MonoDepth (Rey et al., 2022). The other

16 are prior SoTA perspective depth estimators: Metric3D & Metric3Dv2 (Yin et al., 2023; Hu et al., 2024), VGGT (Wang et al., 2025a), MoGe & MoGev2 (Wang et al., 2025c;d), UniDepth & UniDepthv2 (Piccinelli et al., 2024; 2025b), ZoeDepth (Bhat et al., 2023), DepthAnything & DepthAnythingv2 (Yang et al., 2024a;b), and Lotus-D & Lotus-G (He et al., 2024b). These methods are implemented for panoramic scenarios via multi-view splitting and fusion.

**Zero-shot, end-to-end baselines.** Prior zero-shot, end-to-end methods are rare, and their performance are limited by the scarcity of high-quality panoramic depth data. Only 3 methods are compared: UniK3D (Piccinelli et al., 2025a), PanDA (Cao et al., 2025), and DepthAnyCamera (Guo et al., 2025). As evident in Tab. 1, PanDA predicts affine-invariant (relative) depth, while other methods including DA$^2$ predict at least the scale-invariant (biased) depth.

# G DISCUSSIONS ABOUT THE POTENTIAL DISTRIBUTION MISMATCH BETWEEN THE OUT-PAINTER AND TESTING SCENARIOS

The distribution mismatch between training and testing (*i.e.*, applications, experiments, etc.) is a critical issue that may affect the final zero-shot performance. And making the training data more diverse and comprehensive is an effective way to mitigate the distribution mismatch. Thus, we comprehensively select multiple high-quality perspective datasets in different domains (*i.e.*, captured in different scenes with different styles, etc.) as the sources of our panoramic data curation engine. Some examples of the perspective data sources and the curated data are shown in Fig. 11. Also, the training data of the out-painter is also diversely combined from commercially purchased data and self rendered data, in order to fit the diverse domains of multiple perspective data sources. Some examples of the out-painter's training data are shown in Fig. 10.

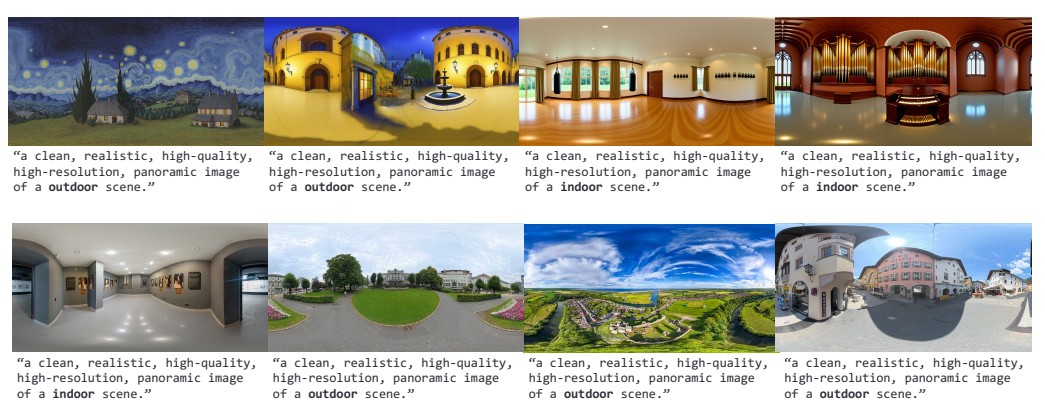

Figure 10: Training data examples of our panoramic our-painter.

# H DISCUSSIONS ABOUT REAL-WORLD PERFORMANCE AS THE TRAINING DATA ARE MOSTLY SYNTHETIC

Real-world performance is surely an important evaluation aspect. We basically use synthetic training data because synthetic data typically exhibit higher quality and sharper edges compared to real-world data, the quality of which may be affected by the many potential limitations of the hardware sensors, *i.e.*, the accuracy issue, resolution issue, and for panoramas, many real-world captures have empty depth values near the poles (please see Fig. 15). Also, it's worthwhile to note that in the adopted three well-recognized benchmarks in panorama depth estimation (please see Tab. 1), two of them (Stanford2D3D (Armeni et al., 2017), Matterport3D (Chang et al., 2017)) are real-world captured, which demonstrate our superior zero-shot generalization ability in real-world scenarios. Additionally, we believe making the training data more diverse and comprehensive is an effective way to mitigate the distribution mismatch, please see Sec. G for more details.

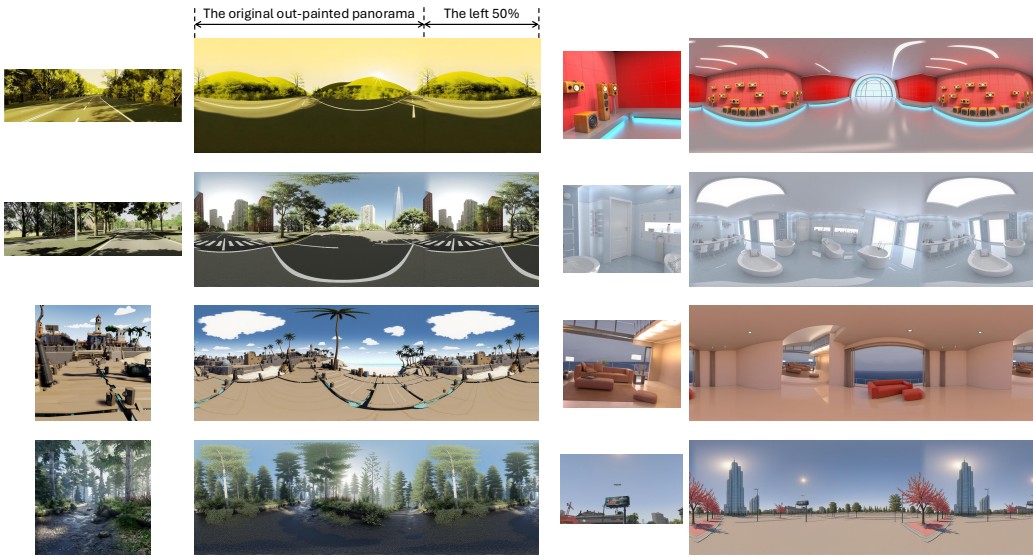

Figure 11: The curated data examples, which come from a different resources with diverse domains. Note that the left 50% of the original panorama is appended on the right, in order to illustrate the horizontal circular consistency.

## I   DISCUSSIONS ABOUT IN-COMPLETE DEPTH SUPERVISION

The performance of DA$^2$ would surely be better if the depth label can also be accurately out-painted, and the missing depth label may decrease the performance especially near the poles. Therefore, we also incorporated a high-quality and native panoramic depth dataset (*i.e.*, Structured3D) with a slightly higher sampling probability than each panoramic transformed perspective dataset (as shown in Tab. 4 in *Supp*'s Sec. B). And while training on this dataset, large errors and subsequently, large gradients near poles can be consistently observed, as illustrated in Fig. 12, which forces the model to predict good distances on these areas.

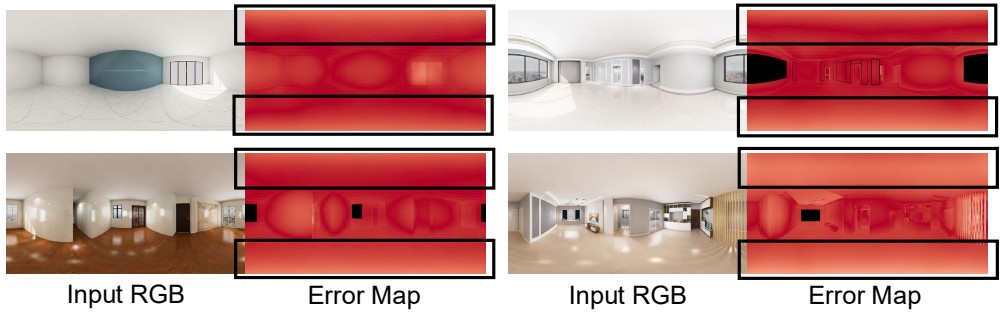

Figure 12: Training supervision visualization (*i.e.*, dense loss maps). Large errors (light color) can be observed particular near the poles, forcing the model to predict good distances on these areas.

In order to better understand the missing annotations at borders (where the depth labels are missing) and centers (where the depth labels are available), we conducted 4 experiments using the native panoramic dataset, Structured3D (Zheng et al., 2020):

1. Using the entire dataset, with all depth labels available, noted as "All".
2. Using the entire dataset, with all depth labels available in the first half and only "centers" depth labels are available in the second half, noted as "Half-center".
3. Using the entire dataset, with all depth labels available in the first half and only "border" depth labels are available in the second half, noted as "Half-border".

4. Using only the first half of the dataset, with all depth labels available, noted as "Half".

We originally expect that "All" is the best and "Half" is the worst due to the difference in scale. However, surprisingly, these 4 groups of experiments have basically the same results as the second row of Tab. 2 (only Structured3D, or S3D, is ✓, which corresponds to "All"). This phenomenon motivates us to more seriously consider the importance of the "diversity" of "domains" in our training data. When the "quantity" is not the "the shortest plank in the barrel", the importance of "diversity" is higher than our expectation. We thus re-analyze the scaling-law curves illustrated in Fig. 2 and Tab. 2. At the beginning, when the number of datasets used is small and the existing diversity is limited, performance grows very rapidly as additional datasets are introduced, since each new dataset contributes significantly to the diversity (*e.g.*, different scene semantics, layout, lighting conditions, etc.). As illustrated, the introduction of Hypersim (Roberts et al., 2021) yields an improvement of 0.97 in AbsRel with only ∼33K's improvement in quantity, the ratio is 0.03 AbsRel(%)/K. The introduction of VKITTI (Cabon et al., 2020) further brings a clear improvement of 0.26 in AbsRel with ∼40K's improvement in quantity, but the ratio is only 0.007 AbsRel(%)/K. As more datasets curated from different sources are introduced, the performance gained from both the diversity and quantity gradually becomes marginal. When the Dynamic Replica (Karaev et al., 2023) was introduced, the gain in AbsRel is only 0.04, though it brings an improvement of ∼290K in quantity.

## J  THE VISUAL QUALITY OF OUT-PAINTED REGIONS

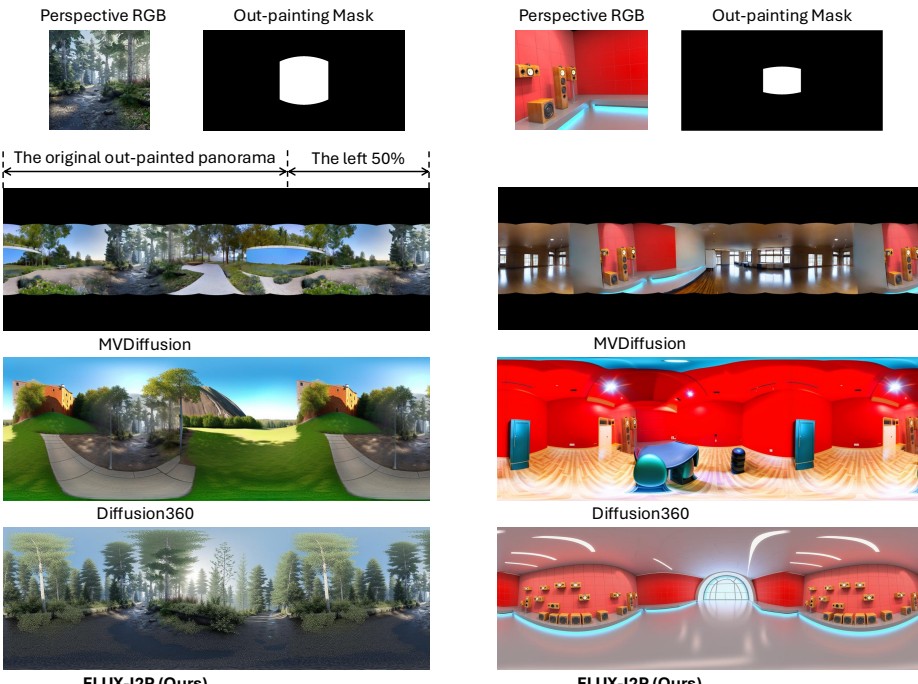

Figure 13: Qualitative comparisons of FLUX-I2P with SoTA baselines. Following Fig. 11, the left 50% is appended on the right of the figure to show the circular consistency.

As an essential part of our panoramic data curation engine, the visual quality of the panoramic out-painter, *i.e.*, the FLUX-I2P is better to be evaluated, which can help in understanding the quality of our curated data. Currently, there exist two well-recognized panoramic out-painters, Diffusion360 (Feng et al., 2023) and MVDiffusion (Tang et al., 2023).

MVDiffusion is capable of generating 7 more perspective images based on a given input perspective image. Combined with the input, these 8 images (each has a horizontal FoV of 90°) constitute a "partial" panorama, with each image separated by a horizontal gap of 45°. However, MVDiffusion assumes the input perspective image has a horizontal FoV of 90° and can not generate the upper and lower poles of the panorama, which is different from our requirement.

Diffusion360's setting aligns with us closer. The inputs of both Diffusion360 and FLUX-I2P are a perspective image and its horizontal FoV (which corresponds to the out-painting mask as shown in Fig. 13). And the output is the out-painted image. In our setting, the input perspective image will firstly be perspective-to-equirectangular (P2E) projected into the spherical space, and be consequently out-painted. During the out-painting, the P2E projected images, which perfectly correspond to the out-painting masks, are completely preserved. However, in Diffusion360's setting, the input perspective image may not be strictly preserved, as evident in Fig. 13.

The qualitative comparisons of FLUX-I2P with Diffusion360 and MVDiffusion is shown in Fig. 13. As illustrated, FLUX-I2P is able to generate panoramas with superior visual quality, high realism, and strong semantic coherence with the input image. While Diffusion360 and MVDiffusion may exhibit clear inconsistencies between their out-painted content and the input image (*e.g.*, Diffusion360's result on the left case, MVDiffusion's result on the right case), exhibit oversaturated style (*e.g.*, Diffusion360's result on the right case).

In order to quantitatively evaluate our FLUX-I2P with SoTA baselines, we conduct a user study across a wide range of evaluation aspects, as shown in Tab. 6 and Fig. 14. This is because there rarely exists a well-recognized benchmark for panoramic image out-painting, where we can calculate metrics like SSIM, LPIPS, PSNR; and it's not straightforward to evaluate the "generation ability" using reconstruction metrics. Also, considering the in-complete panoramic out-painting of MVDiffusion, which largely differs from the settings of FLUX-I2P and Diffusion360, we kindly remove MVDiffusion for the comparison. Four metrics are considered in this quantitative evaluation:

1. The overall visual quality of the panorama (an ERP image) (VQ-ERP).
2. The semantic consistency of the out-painted content and the input image (IN-OUT-CON).
3. The circular seams (SEAM).
4. The visual quality of the panorama under immersive perspective projection (VQ-PER).

Table 6: Winning rate of FLUX-I2P over Diffusion360 measured on 30 cases by human preferences (the higher, the better). A total of 5 people participate (no authors) and 150 comparisons are made. FLUX-I2P consistently out-performs Diffusion360 in the above evaluation perspectives. Please see Fig. 14 for the interface of the user study of one example case.

| Competition | VQ-ERP | IN-OUT-CON | SEAM | VQ-PER |
|---|---|---|---|---|
| FLUX-I2P (Ours) vs. Diffusion360 | 72.7% | 92.7% | 55.3% | 79.3% |

## K  DISCUSSIONS ABOUT SPHEREVIT: PANOFORMER AND SPHEREDEPTH

PanoFormer (Shen et al., 2022) proposed a STLM to initialize the related token positions (replacing conventional ViT positional embedding), using the spatial transformations among ERP domain, spherical domain, and tangent domain. Then, PanoFormer proposed a token flow to "learn" a bias of STLM, to refine it based on image features. SphereDepth (Yan et al., 2022) leverages ISM to represent the spherical structure and use interpolation to transfer the ERP image pixels into (and back from) each sample point on the ISM (controlled by both MR and TR).

Both SphereViT and PanoFormer propose a "sphere-aware" embedding to replace the conventional one. But compared with PanoFormer, SphereViT directly and explicitly inject the entire $360° \times 180°$ spherical structure into the network, while PanoFormer's STLM focuses on the relative positions within a tangent projection patch, which is like a "fine-grained" version of cubemaps.

Both SphereDepth and SphereViT directly and explicitly inject spherical structure into the network. SphereDepth does this even more explicitly as it directly designs sphere network blocks and both the input and prediction are spherical representations sampled on ISM. However, as stated in their original paper: "representing a high-resolution panorama image needs a spherical mesh with high resolution (*i.e.*, high MR) and more triangles.", and "The topological complexity of the spherical mesh exponentially grows with the number of triangles (*i.e.*, MR) increasing, which degrades the computing efficiency and increases the memory footprint.". Also, interpolation is needed to project

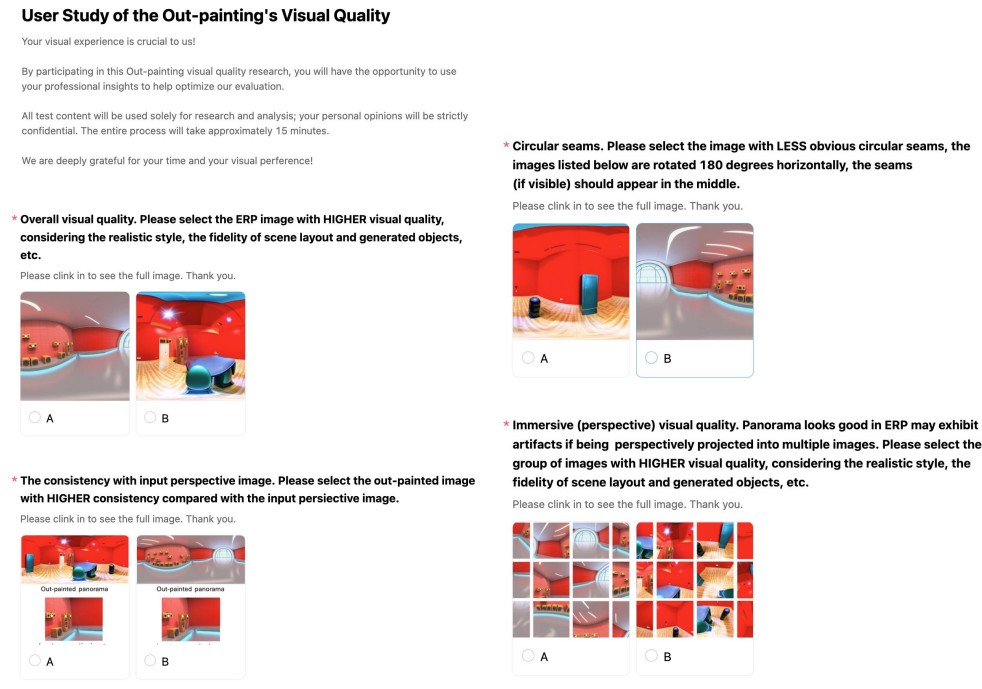

Figure 14: The user study interface of one example case, designed for Tab. 6.

dense pixels into and back from sampled points on ISM. Thus, compared to SphereDepth, SphereViT is more streamlined and may have higher efficiency.

## L   DISCUSSIONS ABOUT SPHEREVIT: S²NET

Also, in order to mitigate the spherical distortions of panoramas, S²Net (Li et al., 2023) utilizes HEALPix (Gorski et al., 2005) as the spherical sampling space. And similar to SphereDepth, S²Net also uses interpolation to map ERP features onto and back from Sphere, formularized as $f_{\pi \to S^2}$ and $f_{S^2 \to \pi}$. S²Net also utilizes cross attention to fuse the spherical features from different fine-grained levels. Compared to S²Net, SphereViT may exhibit higher efficiency because:

1. S²Net fuse features from 4 coarse-to-fine spherical levels via cross-attentions, which may increase the computation burden.

2. Though the HEALPix structure used by S²Net is already optimized for efficiency, it is still much slower than rectangular image features, because its pixels are on a curved sphere instead of a flat plane, and their arrangement is irregular.

3. Feature projections are performed during inference, making it less streamlined.

## M   DISCUSSIONS ABOUT SPHEREVIT: HUSH

### M.1   THE SPHERICAL EMBEDDING AND CROSS-ATTENTION

HUSH (Lee et al., 2025) uses spherical harmonic (SH) basis functions as spherical priors aiming for better feature learning on spherical structures. The core modules are: i. SH-based hierarchical attention module, and ii. SH basis index module. In the first module, SH basis functions serve as geometric priors to guide hierarchical cross-attention between spherical geometry and image features. The second module selects important indexes of SH bases based on the similarity of the task specific feature $f_T$ and embedded SH basis $B_n$, to adaptively emphasize relevant SH bases.

Both HUSH and SphereViT use cross-attention, but the ways how they are using them are different. SphereViT uses spherical embeddings $E_{\text{sphere}}$ as keys and values, and the image feature as queries.

While HUSH, in its SH-based hierarchical attention module, *initially* uses SH feature as query and image feature $f_1$ as key and value. Then, HUSH sequentially and alternatively uses image feature (*e.g.*, $f_2$, $f_3$) or SH feature as query, the output of last cross-attention acts as key and value. HUSH only uses the initial image feature $f_1$ as the key and value, and SH feature as query.

The underlying reason for such differences in injecting spherical awareness is that the structural prior in HUSH (SH coefficients) is learnable but in SphereViT the prior is fixed. SH bases are base functions defined on spherical space. They don't directly represent the spherical structure itself, but are more likely a defined feature space which should also be refined (or learned) with the image features. HUSH also selects different SH bases to accommodate different tasks. While in SphereViT, the spherical embeddings $E_{\text{sphere}}$ directly represent the spherical structure, and thus can be fixed as a global context and be shared across different panoramas. This is similar to the way diffusion models (Rombach et al., 2022) inject users' prompts as global context during image generation.

## M.2 The Improvement from Spherical Embedding

The baseline ViT model does use standard 2D sine-cosine positional embeddings based on the rectangular pixel coordinates (u, v), consistent with conventional ViTs. In contrast, our SphereViT proposes "sphere-aware" spherical embeddings $(\phi, \theta)$ derived from the true azimuth and polar angles of each pixel. This change explicitly encodes the non-uniform sampling of the ERP images and the spherical structure of the $360°$ domain; these are key properties that (u, v) embeddings cannot model. While the overall AbsRel gain may appear moderate, we emphasize that this is because:

1. The introduction of spherical embeddings brings reduced curvature distortion (*e.g.*, more straight walls in Fig. 6 (a)) and improved geometric stability primarily near the poles where the spherical distortions are significant, which may not be fully captured by globally averaged metrics such as AbsRel or RMSE.

2. The panoramas in real-world benchmarks are usually in-complete, missing both the RGB and depth labels near the poles. Some examples are shown in Fig. 15.

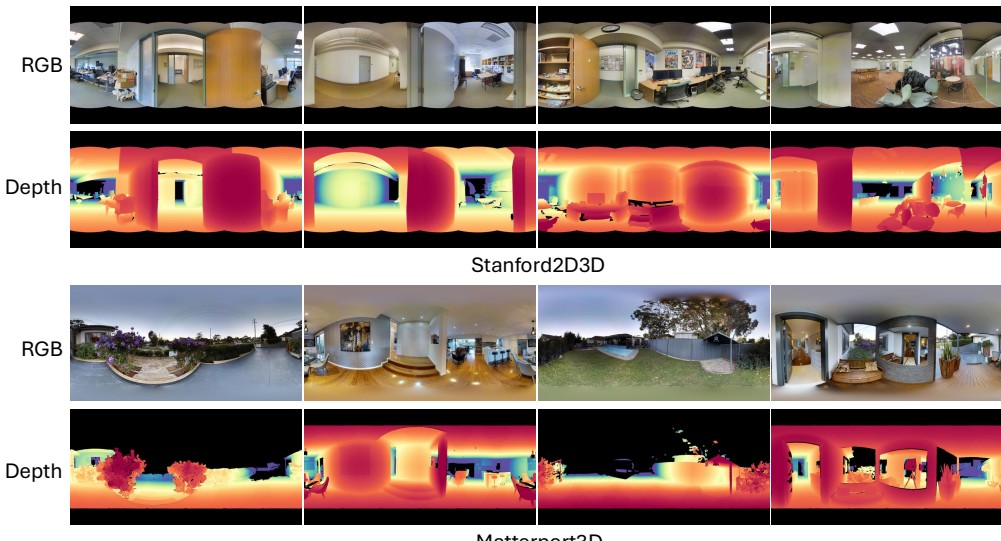

Figure 15: Some RGB and depth examples from the real-world captured benchmarks. Notably, the upper and lower regions of the panoramas are missing, perhaps due to the limitations of hardwares.

## M.3 The Normal Losses

HUSH states in its abstract: *"Finally, by combining the scene features with task-relevant features in the task-specific heads, we perform various scene understanding tasks, including depth, surface normal and room layout estimation.".* While we focus on zero-shot panoramic depth estimation,

because depth estimation can be considered as a feed-forward 3D point cloud reconstruction, which enables a wide range of applications compared to only the 2D visual content.

Both HUSH and SphereViT use normal loss, but the motivations are different. HUSH "performs various scene understanding tasks" (*i.e.*, depth, normal, layout) and its predictions of different tasks come from different prediction heads and different SH bases. Thus, the normal loss (also the depth loss, gradient loss) are the main losses that should be included during training because they directly correspond to the claimed tasks. While SphereViT focuses on depth only, and we consider depth as an important foundational 3D task. The motivation why we use normal loss is aiming to mitigate the artifacts of the reconstructed 3D point cloud (please see Fig. 6) especially near the poles. These artifacted areas generally share similar depth values with GT, yet their normal values differ from GT markedly. Thus, the normal loss of SphereViT is an important auxiliary loss to improve the quality of the predicted depth, and the reconstructed 3D point cloud.

## N  MORE ABLATIONS ABOUT SPHEREVIT'S ARCHITECTURAL DESIGN

In a cross-attention layer, the computation is defined as:

$$\texttt{CrossAttn}\left(Q, K, V\right) = \text{SoftMax}\left(\frac{QW_Q\left(KW_K\right)^\top}{\sqrt{D_k}}\right)\left(VW_V\right), \tag{13}$$

Firstly, the output has the same spatial and temporal dimension as query (Q), that is, the attention's output corresponds strictly with query, where each token incorporates information from key (K) and value (V). This indicates K and V provides "reference information sources" and Q is the "target feature" being updated. Secondly, considering the information flow, the Q determines "where to look", its features generate attention weights with K that decide which parts of V are relevant. The attended information is then aggregated back to Q's positions. Consequently, the data path updates Q while K and V act as conditioning sources that guide the update. Then, during backpropagation the dominant gradient flows toward Q's parameters, since the loss is computed on the attention's output that shares Q's shape. The model thus learns to adjust Q's representations so that they can better extract and integrate information from K,V.

Thus, the Q defines the representation to be enhanced, while the K, V provide context or prior knowledge. This principle underlies SphereViT's design, where image features (as Q) are refined through attention to the spherical embedding (as K, V), enabling the model to absorb geometric priors without modifying the fixed spherical coordinates themselves. Similar design has been well recognized in Stable Diffusion (Rombach et al., 2022), where cross-attention is used to inject users' prompts as global context. During image generation, the image latent features (as Q) are gradually refined through attention to the "fixed" prompt embedding (as K, V), enabling the model to generate images that accurately respond to users' preferences.

Table 7: Ablation studies about SphereViT's architectural design. The **best** and second best performances are bolded and underlined. Following Tab. 6 and 2, the averaged results across multi benchmark datasets are reported. Unit is percentage (%).

| Operations for ablation | AbsRel↓ | RMSE↓ | $\delta_1$ ↑ | $\delta_2$ ↑ |
|---|---|---|---|---|
| Swap Q and K / V in cross-attention | (Model doesn't converge) | | | |
| Replace cross-attention by concatenation and self-attention | 7.42 | 23.04 | 94.31 | 97.98 |
| Replace cross-attention with self-attention | 7.29 | 22.45 | 94.60 | 98.13 |
| Nothing changed | **6.62** | **20.63** | **95.73** | **98.51** |

### N.1  CROSS-ATTENTION VS. SELF-ATTENTION

When the cross-attention module is replaced by self-attention after the addition of the spherical embedding and image features, the model performance also drops notably, as shown in Tab. 7. This is because self-attention treats the spherical embedding "part of" the feature channels, without explicitly establishing any directional interaction between the image features and the geometric prior.

Consequently, the model fails to effectively incorporate spherical structural information, leading to weaker distortion awareness and reduced overall accuracy.

### N.2 CROSS-ATTENTION VS. CONCATENATION AND SELF-ATTENTION

When replacing the cross-attention module with a simple concatenation followed by self-attention, the spherical embedding is merely treated as additional feature channels, without explicitly establishing any directional interaction between the image features and the geometric prior. Similar to Sec. N.1, this operation finally degrades the model's performance, as shown in Tab. 7.

### N.3 SWAPPING QUERY AND KEY / VALUE IN CROSS-ATTENTION

This modification fundamentally changes the direction of "attention flow". As analyzed earlier, in a cross-attention layer, the tokens in Q are the ones being refined, while the K / V tokens act as reference sources. When the roles are swapped, the spherical embedding becomes the Q, the attention tries to update the spherical geometry considering the image features. This disrupts semantic alignment between image features and spherical geometry, preventing the model from absorbing the spherical priors. As a result, the model fails to converge and produces severely degraded performance, confirming that the original attending direction is essential for effective spherical conditioning.

### N.4 MORE ABLATIONS TO SHOW THE COMPREHENSIVE EFFECT OF $E_{\text{SPHERE}}$ AND $\mathcal{L}_{\text{NOR}}$

Table 8: More ablation studies. We here report the quantitative results ***w/*** pano. out-painting, ***w/o*** spherical emb. $E_{\text{sphere}}$, ***w/o*** normal loss $\mathcal{L}_{\text{nor}}$, to better understand the comprehensive effect of $E_{\text{sphere}}$ and $\mathcal{L}_{\text{nor}}$. The last three rows are directly borrowed from Tab. 3.

| Pano. Out-painting | Spherical Emb. $E_{\text{sphere}}$ | Normal Loss $\mathcal{L}_{\text{nor}}$ | AbsRel↓ | RMSE↓ | $\delta_1$ ↑ | $\delta_2$ ↑ |
|:---:|:---:|:---:|:---:|:---:|:---:|:---:|
| ✓ | ✗ | ✗ | 7.27 | 22.48 | 94.77 | 98.19 |
| ✓ | ✗ | ✓ | 6.84 | 20.87 | 95.26 | 98.43 |
| ✓ | ✓ | ✗ | 6.99 | 21.53 | 95.25 | 98.37 |
| ✓ | ✓ | ✓ | **6.62** | **20.63** | **95.73** | **98.51** |

When both the spherical embedding $E_{\text{sphere}}$ and the normal loss $\mathcal{L}_{\text{nor}}$ are removed, as shown in Tab. 8, performance drops significantly. Without $E_{\text{sphere}}$, the model's awareness of spherical geometry will decrease, leading to larger distortions; and without $\mathcal{L}_{\text{nor}}$, local surface smoothness will be less preserved. The results in Tab. 8 further confirm that both components are essential for achieving distortion-aware and high geometrical fidelity predictions.

## O METRIC DEPTH VERSION OF DA$^2$

Table 9: Quantitative comparisons on metric depth estimation, compared with UniK3D.

| Method | Stanford2D3D | | | | Matterport3D | | | | PanoSUNCG | | | |
|:---|:---:|:---:|:---:|:---:|:---:|:---:|:---:|:---:|:---:|:---:|:---:|:---:|
| | AbsRel↓ | RMSE↓ | $\delta_1$ ↑ | $\delta_2$ ↑ | AbsRel↓ | RMSE↓ | $\delta_1$ ↑ | $\delta_2$ ↑ | AbsRel↓ | RMSE↓ | $\delta_1$ ↑ | $\delta_2$ ↑ |
| UniK3D | 17.95 | 48.50 | 78.23 | 86.63 | 22.24 | 66.80 | 66.34 | 92.81 | 12.60 | 27.20 | 88.90 | 98.47 |
| DA$^2$ | **16.56** | **45.73** | **80.04** | **90.32** | **17.53** | **60.85** | **68.21** | **96.13** | **11.23** | **21.03** | **89.20** | **98.53** |

We sincerely agree that metric depth estimation can expand the downstream applications. The current version of this work only supports "scale-invariant" or "biased" because the authors consider the "metric-scale" a little ill-posed because sometimes the metric-scale can't be reflected in the panoramas. For example, if the entire scene is scaled equal-proportionally and linearly, the 360° pictures captured will still remain the same. However, these situations are rare. Thus, we also train a metric version of DA$^2$ by directly supervising the model with metric GT depth, to explore its metric depth performance. As illustrated in Tab. 9, thanks to the data curation engine and the SphereViT which is "sphere-aware", DA$^2$ consistently achieves higher performance compared with UniK3D.

# P    DISCUSSIONS ABOUT THE LEFT-RIGHT SEAMS AND POTENTIAL SOLUTIONS

We sincerely acknowledge that our model may exhibit left-right seams in the predicted distance maps and the reconstructed 3D. The seam artifacts mainly arise from two factors:

1. The input ERP images introduce artificial left-right boundaries. Commonly-used neural network architectures can not easily enforce this horizontal consistency. Because there is a lack of adjacent correlation between the left-right boundaries which should be adjacent but are separated in ERP images.

2. The out-painted RGB may include slight color or structure discontinuities at the left-right boundaries, which may also influence the horizontal consistency of SphereViT.

For potential solutions, performing a horizontal circular rotating augmentation may be an effective way to enforce the left-right boundary consistency. Specifically, for if the panoramas in the training data have perfectly aligned left-right boundaries and the depth maps are complete and also perfectly aligned at left-right boundaries. Then, during training, we can augment each data sample into multiple ones ($1 \rightarrow N$), and for each augmented sample (the $i^{\text{th}}$ sample), we horizontally rotate the panorama and the depth map with $360° \cdot \frac{i}{N}$. We believe adding such a horizontal circular rotating augmentation will mitigate the left-right seams better than adding a simple smoothing interpolation near the left-right seams on the predicted depth maps, which we have tried but it didn't work well.

# Q    MORE QUALITATIVE RESULTS COMPARED WITH UNIK3D AND MOGEV2

We here provided more qualitative results compared with SoTA baselines: UniK3D (Piccinelli et al., 2025a) and MoGev2 (Wang et al., 2025d). Consistent with Fig. 5, compared with UniK3D and MoGev2, DA$^2$ delivers more accurate geometric predictions. MoGev2 divides the panorama into several perspective patches, and performs perspective depth estimation on each patch, then these perspective depth patches are fused into a panorama. It takes much longer time during inference and its quality is limited by the geometrical consistency of the estimated depth at each patch. Such inconsistencies usually take place near the poles in our practice, perhaps due to the RGB inputs of MoGev2's training data rarely "look up" or "down". As illustrated in Fig. 16, MoGev2 may exhibit irregularly deformed or distorted 3D geometry.

UniK3D aims to achieve robust generalization across different cameras through training on large-scale datasets. However, due to the limited amount of panoramic depth data in its training data (only $\sim$29K), UniK3D's performance on panoramic images, particularly near the poles where spherical distortion becomes significant, remains clearly constrained. Its performance near the equator is relatively strong, benefiting from the smaller domain gap to conventional perspective images. Also, in its official demos (the `equirectangular.jpg` and `venice.jpg` under this folder), the top and bottom $\sim$10% of the panoramas are cropped to ensure reliable results. As shown in Fig. 16, UniK3D exhibits noticeably inferior performance near the poles, producing clearly distorted geometry in regions such as ceilings, floors, and the upper and lower regions of side walls.

We here primarily show the qualitative comparisons on indoor scenes because indoor scenes can more clearly demonstrate our superior geometrical fidelity. Due to the much larger space represented in outdoor panoramas, the lower regions of panoramas usually represent a very small area of ground, which only covers a tiny portion in the whole 3D scene. And the upper regions are usually part of the sky. As discussed above, the inconsistencies between patches in MoGev2 usually happen near the poles, and the UniK3D's performance on panoramic images also typically fail short near the poles. Thus, outdoor scenes may make it more difficult in very clearly demonstrating the superiority of DA$^2$'s geometrical estimation, compared with indoor scenes.

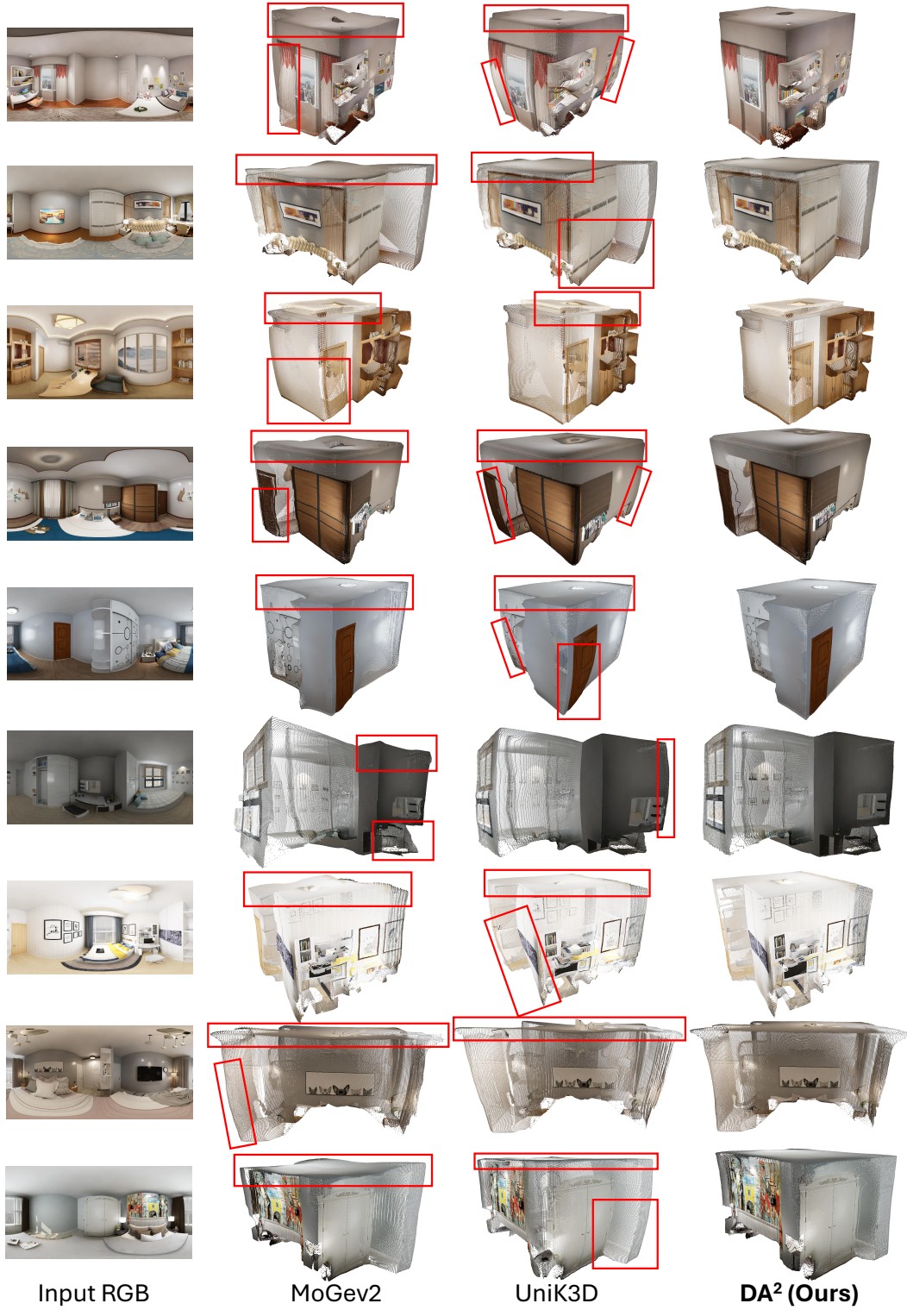

Input RGB          MoGev2          UniK3D          **DA² (Ours)**

Figure 16: More qualitative results compared with UniK3D and MoGev2. Consistent with Fig. 5, DA² still delivers more accurate geometric predictions. Please zoom-in for more details.

