# OpenReview forum: "DA$^{2}$: Depth Anything in Any Direction"
_ICLR.cc/2026/Conference — ICLR 2026 Poster_

### Official Review · Reviewer_RcRx · 2025-10-28

**Soundness:** 3
**Presentation:** 3
**Contribution:** 3
**Rating:** 8
**Confidence:** 5

**Summary:**

This paper proposes DA$^2$, an end-to-end, panoramic depth estimation model that achieves strong zero-shot generalization.

Previous methods face challenges in zero-shot panoramic depth estimation because:
1. [Data scarcity] Existing panoramic depth datasets are rare
2. [Spherical distortions] Common equirectangular projection of panoramas clearly distorts near poles.

Correspondingly, the authors addess this problems with two main contributions:
1. [Panoramic data curation engine] Considering the rich data in perspective depth estimation, this paper proposes a panoramic data curation engine. Firstly, the perspective RGB / depth images are transformed into spherical space via Perspective-to-Equirectangular (P2E) projection. Then, for the transformed RGB is panoramic out-painted to obtain the "full" panorama. This engine curates additional 540K data samples.
2. [SphereViT] A ViT network that explicitly incorporates spherical coordinates into latent feature via spherical embeddings, designed to mitigates the effect of spherical distortions.

To validate DA$^2$, the authors conduct a comprehensive comparison, including both panoramic depth estimation methods and perspective methods. The results clearly show the SOTA performance of DA$^2$.

**Strengths:**

1. Reasonable analysis of existing methods, and straightforward solutions targeting the existing limitations. The existing issues (Data scarcity, Spherical distortions) is critical in panoramic depth estimation. And the solutions (Panoramic data curation engine, SphereViT) effectively address the limitations, making this paper valuable in panoramic depth estimation.

2. In the experiments, the author built a comprehensive benchmark among various datasets, and compared DA$^2$ with both zero-shot / in-domain, panoramic / perspective approaches. The quantitative and qualitative results clearly show DA$^2$'s SOTA performance. The ablation studies clearly show the performance gained via scaling-up with perspective data, also the spherical embeddings in SphereViT.

3. The authors claimed that the code and the curated data will be open-sourced. This will be a valuable contribution to the research community.

4. The writing is clear and also the figures. The paper is easy to follow.

**Weaknesses:**

1. Performance concerns on real-world panoramas: The reviewer noticed that the curated panoramic data are basically come from synthetic perspective data. The domain gap between synthetic data and real-world data may decrease the model's performance on real-world panoramas.

2. While the data curation engine is effective, the depth label is largely missing, which is better obtained. The author should add more discussions on this matter.

3. More qualitative results: There are only 2 cases in this paper's qualitative comparison.

4. The reviewer also noticed left-right seams in the reconstructed 3D point clouds. The author should add more discussions about this problem, which can be critical in real-world applications.

5. Minors:

    (1). The arrow on the right figure of figure 2 is in opposite direction.

    (2). The citation of MVS-Synth in line 371, use \citep instead.

**Questions:**

Please refer to weaknesses.

---

> ### Author Response · Authors · 2025-11-20
> **Author's response to reviewer RcRx**
>
> **Response to weaknesses & questions:**
>
> > **1.** Performance concerns on real-world panoramas as the training datasets are basically synthetic.
>
> We sincerely acknowledge that real-world performance is surely an important evaluation aspect. We basically use synthetic training data because synthetic data typically exhibit higher quality and sharper edges compared to real-world data, the quality of which may be affected by the many potential limitations of the hardware sensors, i.e., the accuracy issue, resolution issue, and for panoramas, many real-world captures have empty depth values near the poles (please see *Supp*’s Fig. 15). Also, it’s worthwhile to note that in the adopted three well-recognized benchmarks in panorama depth estimation (please see paper’s Tab. 1), two of them are real-world captured, which demonstrate our superior zero-shot generalization ability in real-world scenarios. Additionally, we believe making the training data more diverse and comprehensive is an effective way to mitigate the distribution mismatch, to avoid repetition, please see *Supp*’s Sec. G and our response to reviewer t7tq (point 1) for more details.
>
> > **2.** More discussions are needed about the missing depth labels in the curated datasets.
>
> We also kindly agree that the performance of DA$^{2}$ would surely be better if the depth label can also be accurately out-painted, and the missing depth label may decrease the performance especially near the poles. To avoid repetition, please see *Supp*’s Sec. I and our response to reviewer GN9K (point 1.1) for our response to this concern. Thank you for your understanding.
>
> > **3.** More qualitative results are needed.
>
> Thanks for your suggestion. We have provided more qualitative results compared with SoTA baselines in our *Supp*’s Fig. 16. Consistent with paper’s Fig. 5, compared with UniK3D[1] and MoGev2[2], DA$^2$ delivers more accurate geometric predictions.
>
> MoGev2[1] divides the panorama into several perspective patches, and performs perspective depth estimation on each patch, then these perspective depth patches are fused into a panorama. It takes much longer time during inference and its quality is limited by the geometrical consistency of the estimated depth at each patch. Such inconsistencies usually take place near the poles in our practice, perhaps due to the RGB inputs of MoGev2’s training data rarely “look up” or “down”. As illustrated in *Supp*’s Fig. 16, MoGev2 may exhibit irregularly deformed or distorted 3D geometry.
>
> UniK3D[2] aims to achieve robust generalization across different cameras through training on large-scale datasets. However, due to the limited amount of panoramic depth data in its training data (only $\sim$29K), UniK3D’s performance on panoramic images, particularly near the poles where spherical distortion becomes significant, remains clearly constrained. Its performance near the equator is relatively strong, benefiting from the smaller domain gap to conventional perspective images. Also, in its official demos (the `equirectangular.jpg` and `venice.jpg` under [this folder](https://github.com/lpiccinelli-eth/UniK3D/tree/main/assets/demo)), the top and bottom $\sim$10% of the panoramas are cropped to ensure reliable results. As shown in *Supp*’s Fig. 16, UniK3D exhibits noticeably inferior performance near the poles, producing clearly distorted geometry in regions such as ceilings, floors, and the upper and lower regions of side walls.
>
> We here primarily show the qualitative comparisons on indoor scenes because indoor scenes can more clearly demonstrate our superior geometrical fidelity. Due to the much larger space represented in outdoor panoramas, the lower regions of panoramas usually represent a very small area of ground, which only covers a tiny portion in the whole 3D scene. And the upper regions are usually part of the sky. As discussed above, the inconsistencies between patches in MoGev2 usually happen near the poles, and the UniK3D’s performance on panoramic images also typically fail short near the poles. Thus, outdoor scenes may make it more difficult in very clearly demonstrating the superiority of DA$^2$’s geometrical estimation, compared with indoor scenes.
>
> This discussion has been updated in PDF (*Supp*’s Sec. Q).
>
>
> [1] Ruicheng Wang, et al. MoGe-2: Accurate Monocular Geometry with Metric Scale and Sharp Details. NeurIPS 2025.
>
> [2] Luigi Piccinelli, et al. UniK3D: Universal Camera Monocular 3D Estimation. CVPR 2025.
>
> > **4.** More discussions are needed regarding the left-right seams in the reconstructed 3D point clouds.
>
> To avoid repetition, please see our *Supp*’s Sec. P and our responses to reviewer xeLR (point 3) for more details. We are grateful for your understanding.
>
> > **5.** Typos
>
> Thanks for pointing out and your in-depth reading of our submission, we have updated the PDF correspondingly.

---

> > ### Comment · Reviewer_RcRx · 2025-11-22
> >
> > Thanks for the reply which has addressed my previous concerns. I have no further questions.

---

### Official Review · Reviewer_xeLR · 2025-10-29

**Soundness:** 3
**Presentation:** 3
**Contribution:** 3
**Rating:** 6
**Confidence:** 4

**Summary:**

This paper introduces DA2, a method for zero-shot, monocular panoramic depth estimation. The work tackles two fundamental challenges in this domain: the scarcity of large-scale panoramic training data and the geometric distortions inherent in the equirectangular projection.

To address data scarcity, the paper proposes a panoramic data curation engine. This engine leverages abundant perspective RGB-D datasets by first projecting a perspective image into a partial panorama and then using a image generative model (i.e. FLUX-I2P) to out-paint the partial panorama into a full one. This process increased their training set size by nearly tenfold, from ~63K to ~607K pairs.

To handle spherical distortions, the paper proposes SphereViT, a ViT-based architecture. Instead of using standard 2D positional embeddings, SphereViT creates a fixed Spherical Embedding derived from the spherical coordinates (azimuth and polar angles) of each image patch. It then uses cross-attention, with the image features as queries and the spherical embedding as keys and values, to explicitly inject distortion-awareness into the feature representations.

Through extensive experiments on standard benchmarks (Stanford2D3D, Matterport3D, PanoSUNCG), DA2 is shown to achieve state-of-the-art performance, significantly outperforming prior zero-shot methods and many specialized in-domain models with high inference efficiency.

**Strengths:**

The paper's primary strength is its two-fold innovation.

1) The data curation engine creatively uses a generative model (i.e. FLUX-I2P) to synthesize panoramic training data from abundant perspective data, the authors have effectively broken through the data barrier in the academic community that restricts a open source panoramic depth estimation comparable to commercial models. The resulting "scaling law" (Fig. 2) is a powerful demonstration of the data-centric approach.

2) The SphereViT architecture introduces a neat and elegant solution to the problem of spherical distortion. Its use of cross-attention with a fixed spherical embedding is a more principled and efficient approach than the fusion strategies common in prior work.

The resulting model, DA2, is not only accurate but also efficient. As a fully end-to-end method, it is significantly faster than fusion-based alternatives, making it practical for real-world applications, as mentioned in the appendix. The planned release of the code and the large-scale curated panoramic dataset is a big contribution in itself and will be a substantial asset to the research community.

The paper is easy to follow. The experimental validation is thorough, featuring a comprehensive benchmark against a large number of competing methods.

**Weaknesses:**

The paper is novel and effective, the following points are intended as constructive suggestions for further improvement.

1) About Out-painter

The quality of the curated data, and thus the performance of DA2, is intrinsically linked to the performance of the FLUX-I2P out-painter. Any biases or artifacts of the generative model could be implicitly learned by DA2. A brief discussion on the limitations of the out-painter, and how different out-painters affect depth estimation would add valuable context.

2) About SphereViT

The cross-attention mechanism in SphereViT is a key architectural novelty. The ablation study shows that including the spherical embedding is crucial. However, it does not compare the proposed cross-attention design against other plausible alternatives for incorporating this embedding. For instance, the standard ViT approach (adding the embedding to image features before self-attention) or simply concatenating the spherical coordinates as extra channels would provide stronger evidence that the specific cross-attention is optimal.

3) About Seam Artifact

The paper acknowledges visible seams at the panorama's left-right boundary as a limitation. Given that SphereViT is designed with explicit knowledge of spherical coordinates, it is somewhat surprising that this issue persists. A brief discussion on why the current architecture does not fully resolve this and potential solutions would strengthen the paper.

**Questions:**

Thank you for this good work. Besides the points raised in the discussion of weakness, I have a few further questions to better understand the nuances of your contributions.

1) Current image generative methods are capable of out-painting depth information. If the perspective depth maps are also out-painted, would this lead to performance improvements?

2) As mentioned in the paper, the depth estimation model and data will be open-sourced. Will the FLUX-I2P data and model also be open-sourced? The paper does not provide sufficient details about the data used to train FLUX-I2P.

3) Please discuss the relevance to the method proposed in "S2Net: Accurate Panorama Depth Estimation on Spherical Surface". If relevant, please cite this work.

4) In Table 3, what would the results be if only Pano. out-painting is used without SphereViT and Normal loss?

5) In Section A.1, the authors mention that only translation is needed to align the 3D points from panoramas taken at different locations. Please explain why rotation is not needed?

6) Please proofread the manuscript again to identify and correct spelling errors—for example, "date" in Line 380.

---

> ### Author Response · Authors · 2025-11-20
> **Author's response to reviewer xeLR 1/2**
>
> **Response to weaknesses:**
>
> > **1.** More discussions on the limitations of the out-painter, and how different out-painters affect depth estimation would add valuable context.
>
> We sincerely agree that as an essential part of our panoramic data curation engine, the quality of the panoramic out-painter ,i.e., the FLUX-I2P is better to be deeply analyzed and evaluated, which can help in understanding the quality of our curated data. As shown in our *Supp*’s Fig. 11, though our FLUX-I2P is able to generate high-fidelity panoramas with strong coherence with perspective images to be curated, FLUX-I2P does have some limitations. For instance, the messy layout of the out-painted scene (as shown in *Supp*’s Fig. 11, second column, second row) and the left-right seams sometime appear in the out-painted panorama (as shown in *Supp*’s Fig. 11, first column, second row).
>
> We also follow your suggestions to compare our FLUX-I2P’s out-painting quality with SoTA panoramic out-painters, e.g., Diffusion360[1] and MVDiffusion[2]. To avoid repetition, please see our *Supp*’s Sec. J and our response to reviewer GN9K (point 1.2)  for the detailed comparisons and analysis.
>
> [1] Mengyang Feng, et al. Diffusion360: Seamless 360 Degree Panoramic Image Generation based on Diffusion Models. arXiv 2023.
>
> [2] Shitao Tang, et al. MVDiffusion: Enabling Holistic Multi-view Image Generation with Correspondence-Aware Diffusion. NeurIPS 2023.
>
> > **2.** More ablations on the cross-attention mechanism in SphereViT
>
> For cross-attention vs. self-attention, please see our *Supp*’s Sec. N and N.1 and our response to reviewer GN9K (point 2.2). For cross-attention vs. concatenation, please see our *Supp*’s Sec. N and N.3. We are grateful for your understanding in avoiding repetition.
>
> > **3.** Why the current architecture does not fully resolve the seam artifact and potential solutions.
>
> We sincerely acknowledge that our model may exhibit left-right seams in the predicted distance maps and the reconstructed 3D. The seam artifacts mainly arise from two factors:
>
> i. The input ERP images introduce artificial left-right boundaries. Commonly-used neural network architectures can not easily enforce this horizontal consistency. Because there is a lack of adjacent correlation between the left-right boundaries which should be adjacent but are separated in ERP images.
> ii. The out-painted RGB may include slight color or structure discontinuities at the left-right boundaries, which may also influence the horizontal consistency of SphereViT.
>
> For potential solutions, performing a horizontal circular rotating augmentation may be an effective way to enforce the left-right boundary consistency. Specifically, for if the panoramas in the training data have perfectly aligned left-right boundaries and the depth maps are complete and also perfectly aligned at left-right boundaries. Then, during training, we can augment each data sample into multiple ones ($1\rightarrow N$), and for each augmented sample (the $i^\text{th}$ sample), we horizontally rotate the panorama and the depth map with $360^\circ\cdot \frac{i}{N}$. We believe adding such a horizontal circular rotating augmentation will mitigate the left-right seams better than adding a simple smoothing interpolation near the left-right seams on the predicted depth maps, which we have tried but it didn’t work well.
>
> This discussion has been updated in PDF (*Supp*’s Sec. P).

---

> ### Author Response · Authors · 2025-11-20
> **Author's response to reviewer xeLR 2/2**
>
> **Response to questions:**
>
> > **1.** If the perspective depth maps are also out-painted, would this lead to performance improvements?
>
> We also kindly agree that the performance of DA$^{2}$ would surely be better if the depth label can also be accurately out-painted, and the missing depth label may decrease the performance especially near the poles. Please see *Supp*’s Sec. I and our response to reviewer GN9K (point 1.1) for our response to this concern, thank you for your understanding in avoiding repetition.
>
> > **2.** The releasing of FLUX-I2P
>
> We sincerely thank you for your interest in the FLUX-I2P model. Yes, we will release both the checkpoint and the inference code of FLUX-I2P right after the acceptance of this paper. The training data for FLUX-I2P is proprietary to the company and is not allowed for release.
>
> > **3.** Discussions with S$^2$Net [3] and cite this work.
>
> Thank you for mentioning S$^2$Net [3]. We sincerely apologize for our unintentional unawareness of S2Net. Please see our discussion with S2Net below.
>
> Also, in order to mitigate the spherical distortions of panoramas, S$^2$Net utilizes HEALPix[4] as the spherical sampling space. And similar to SphereDepth, S$^2$Net also uses interpolation to map ERP features onto and back from Sphere, formularized as $f_{\pi\rightarrow S^2}$ and $f_{S^2\rightarrow \pi}$. S$^2$Net also utilizes cross attention to fuse the spherical features from different fine-grained levels. Compared to S$^2$Net, SphereViT may exhibit higher efficiency because:
> i. S$^2$Net fuse features from 4 coarse-to-fine spherical levels via cross-attentions, which may increase the computation burden.
> ii. Though the HEALPix structure used by S$^2$Net is already optimized for efficiency, it is still much slower than rectangular image features, because its pixels are on a curved sphere instead of a flat plane, and their arrangement is irregular.
> iii. Feature projections are performed during inference, making it less streamlined.
>
> [3] Meng Li, et al. S2Net: Accurate Panorama Depth Estimation on Spherical Surface. RAL 2023.
>
> [4] K. M. Gorski, et al. HEALPix -- a Framework for High Resolution Discretization, and Fast Analysis of Data Distributed on the Sphere. The Astrophysical Journal, Volume 622, Number 2, 2005.
>
> > **4.** Ablation results w/ only the pano. out-painting, w/o SphereViT and w/o normal loss.
>
> When both the spherical embedding $E_\text{sphere}$ and the normal loss $L_\text{nor}$ are removed, as shown in paper’s Tab. 3, performance drops significantly. Without $E_\text{sphere}$, the model's awareness of spherical geometry will decrease, leading to larger distortions; and without ${L}_\text{nor}$, local surface smoothness will be less preserved. The results in *Supp*’s Tab. 8 (or the table below) further confirm that both components are essential for achieving distortion-aware and high geometrical fidelity predictions.
>
> Table: More ablation studies. We report the quantitative results **w/** pano. out-painting, **w/o** spherical embedding $E_\text{sphere}$, and **w/o** normal loss $L_\text{nor}$, to better understand the comprehensive effect of $E_\text{sphere}$ and $L_\text{nor}$. Unit is percentage (%).
>
> | Pano. Out-painting | Spherical Emb. $E_\text{sphere}$ | Normal Loss $L_\text{nor}$ | AbsRel ↓ | RMSE ↓ | δ₁ ↑ | δ₂ ↑ |
> |:--:|:--:|:--:|:--:|:--:|:--:|:--:|
> | ✅ | ❌ | ❌ | 7.27 | 22.48 | 94.77 | 98.19 |
> | ✅ | ❌ | ✅ | _6.84_ | _20.87_ | _95.26_ | _98.43_ |
> | ✅ | ✅ | ❌ | 6.99 | 21.53 | 95.25 | 98.37 |
> | ✅ | ✅ | ✅ | **6.62** | **20.63** | **95.73** | **98.51** |
>
> This discussion has been updated in PDF (*Supp*’s Sec. N.4).
>
> > **5.** Why is rotation not needed in the multi-panorama reconstruction application?
>
> Thanks for pointing out and for your interest in our applications. Of course, thanks to the robustness and high-quality of our panoramic depth estimator, scaling is not needed for alignment. For example, the height of the reconstructed rooms are nearly the same. And in our cases, the rotations are not needed because
>
> i. The Z-directions of these 360$^\circ$ cameras are vertical to the ground.
> ii. The directions from the cameras to the central pixel of the multi view panoramas are facing the same direction in the space where the multi view panoramas are captured. For example, panorama A and B have a wall and a door overlapped, if the door is located at the left behind A's camera, then it will also be at the right front of B’s camera.
>
> This discussion has been updated in PDF (*Supp*’s Sec. A.1, second paragraph).
>
> > **6.** Typos
>
> Thanks for pointing out and your in-depth reading of our submission, we have updated the PDF correspondingly.

---

> > ### Comment · Reviewer_xeLR · 2025-11-28
> >
> > After reading all the reviews and rebuttals, most of my concerns have been resolved. I’ll maintain my original positive rating, as the work offers valuable insights for the community, and I support its acceptance.

---

### Official Review · Reviewer_c7zZ · 2025-10-31

**Soundness:** 3
**Presentation:** 4
**Contribution:** 2
**Rating:** 6
**Confidence:** 3

**Summary:**

This paper aims to create a universal, end-to-end depth estimation model that generalizes across domains and any directions for panoramic application.
Main contribution:
1. provide a synthetic panoramic data engine/pipeline mainly powered by utilizing FLUX-I2P out painting. And evaluated interest scaling law on that curated data.
2. Design a SphereViT which incorporates spherical embedding, can yield accurate geometrical estimation by distortion-aware image features.
3. Evaluation on zero-shot/in-domain benchmark shows strong quality.

**Strengths:**

1. Good work which represents a step toward universal depth perception across all directions.

2. It combines synthetic panoramic data and sphere-aware architecture could inspire like Geometry-consistent generative modeling.

3. Solid results on comprehensive panoramic benchmarks show good zero-shot performance, outperform in-domain models proved good generalizability.

**Weaknesses:**

1. The large-scale training set is created by FLUX I2P. This work tries to find a good way to utilize the strong FLUX base model capacity. However, the generation pipeline might have distribution gaps like lighting environment & geometry consistency may still have some gaps from real panoramic scenes.
2. Efficiency: this paper’s E2E pipeline is faster than fusion based method but remains at same level with other E2E methods like UniK3D. Both are 0.3s. The E2E pipeline seems not novel.
3. The data and ViT arch design are mainly focused on depth estimation quality improvement. Other methods like PanDa, DepthAnyCamera or UniK3D could also cover the tasks.

**Questions:**

1. What if we collected more real panoramic data (not limited in some domains) and use them to evaluate the scaling law? It should also have improvement for depth estimation task.
2. Can we also use video-generation model to generate some temporal panoramic data for temporal consistency? Single frame is easier to handle.
3. For the ablation study, spherical embedding didn’t show too much improvement (less than 1 point), is there any other better design for the specific utilization of spherical information?

---

> ### Author Response · Authors · 2025-11-20
> **Author's response to reviewer c7zZ 1/2**
>
> **Response to weaknesses:**
>
> > **1.** The potential distribution gap (like lighting environment & geometry consistency) between the out-painter and the real panoramic scenes.
>
> In our perspectives, the distribution mismatch between training and testing (applications, experiments) is a critical issue that may affect the final zero-shot performance. To avoid repetition, please see *Supp*’s Sec. G and our response to reviewer t7tq (point 1) for more details. We sincerely thank you for your understanding.
>
> > **2.** Efficiency: this paper’s E2E pipeline is faster than fusion-based methods but remains at the same level with other E2E methods like UniK3D. The E2E pipeline seems not novel.
>
> Yes. We sincerely agree with your opinion. And as stated in line 130-139, both the E2E design and the higher efficiency compared with fusion-based methods are not listed as our key contributions. Also, regarding the efficiency reported in Fig. 5 of the paper, the 0.3 s runtime was measured on an image of size 1024$\times$2048. We would like to revise this to 0.1 s, which was tested on a 512$\times$1024 image, consistent with the test setting described in line 959-960. MoGev2 resizes each perspective patch into 512$\times$512 before depth estimation, its running time won't be affected by the resolution of the input panorama.
>
> > **3.** The data and ViT arch design are mainly focused on depth estimation quality improvement. Other methods like PanDa[1], DepthAnyCamera[2] or UniK3D[3] could also cover the tasks.
>
> We sincerely acknowledge that PanDa[1], DepthAnyCamera[2], and UniK3D[3] are important prior works in panoramic depth estimation and they’ve made significant progress.
>
> PanDA employs Möbius transformation–based data augmentation for self-supervised panoramic depth learning. By introducing geometric transformations that preserve the topology of the sphere, PanDA effectively expands the scale of training data and enhances cross-view consistency. However, its training data include only around 20K labeled and 92K unlabeled panoramic images, which limits the overall diversity and scale of supervision. Thus, PanDA’s performance may remain suboptimal.
>
> DepthAnyCamera seeks to generalize depth estimation across various camera types by projecting perspective images with diverse XFoV (20$^\circ-$124$^\circ$, $\ll$360$^\circ$) into spherical coordinates. This design allows the model to handle a wide range of perspective cameras under a unified representation space. Nevertheless, because the training data do not cover full-view panoramas, the model’s performance still remains constrained.
>
> UniK3D aims to achieve robust generalization across different cameras through training on large-scale datasets. However, due to the limited amount of panoramic depth data in its training data (only $\sim$29K), UniK3D’s performance on panoramic images, particularly near the poles where spherical distortion becomes significant, still remains constrained (please see paper’s Fig. 5 and *Supp*’s Fig. 16). Its performance near the equator is relatively strong, benefiting from the smaller domain gap to conventional perspective images.
>
> [1] Zidong Cao, et al. PanDA: Towards Panoramic Depth Anything with Unlabeled Panoramas and Mobius Spatial Augmentation. CVPR 2025.
>
> [2] Yuliang Guo, et al. Depth Any Camera: Zero-Shot Metric Depth Estimation from Any Camera. CVPR 2025.
>
> [3] Luigi Piccinelli, et al. UniK3D: Universal Camera Monocular 3D Estimation. CVPR 2025.

---

> ### Author Response · Authors · 2025-11-20
> **Author's response to reviewer c7zZ 2/2**
>
> **Response to questions:**
>
> > **1.** What if we collect more real panoramic data (not limited to some domains) and use them to evaluate the scaling law? It should also have improvement for depth estimation.
>
> Yes. Collecting more real world data is absolutely beneficial once the sensors are at high-quality, that is, the panoramas should be complete (not partial like our *Supp*’s Fig. 15), and the depth labels should be accurate and dense. Currently, it is not so hard to capture high-quality panoramic RGB images given a 360 camera, but to the best of our knowledge, there exists nearly no accurate and dense 360 depth sensors. Developing accurate and dense 360 depth sensors are critical for the future development of immersive computer vision.
>
> > **2.** Can we also use a video-generation model to generate some temporal panoramic data for temporal consistency? Single frames are easier to handle.
>
> We appreciate your inspirable question. Yes, we sincerely agree that if there exists high-quality panoramic videos with GT camera trajectories, measuring the temporal consistency on temporal panoramic inputs would be a valuable perspective to evaluate the robustness of our model. Such measurements can also be used as self-supervision sources and have been studied in prior works like CVD[4] and Robust-CVD[5]. And we also agree that single frames are easier to handle. However, for panoramic video generation models, we kindly argue they’re better be leveraged as part of the data curation engines to scale-up the panoramic data; or as the base model architecture for panoramic video perception tasks.
>
> [4] Xuan Luo, et al. Consistent Video Depth Estimation. SIGGRAPH 2020.
>
> [5] Johannes Kopf, et al. Robust Consistent Video Depth Estimation. CVPR 2021.
>
> > **3.** For the ablation study, spherical embedding didn’t show too much improvement (less than 1 point), is there any other better design for the specific utilization of spherical information?
>
> We thank you for your interest in SphereViT and your in-depth reading. To avoid repetition, please check our *Supp*’s Sec. M.2 and our responses to reviewer GN9K (point 2.1) for our response to this question. For potential better designs, performing a horizontal circular rotating augmentation may further improve the “spherical awareness”. Also, to avoid repetition, please see our *Supp*’s Sec. P and our responses to reviewer xeLR (point 3) for more details. Thank you for your understanding.

---

### Official Review · Reviewer_GN9K · 2025-11-01

**Soundness:** 2
**Presentation:** 4
**Contribution:** 3
**Rating:** 6
**Confidence:** 4

**Summary:**

This paper aims to build a strong zero-shot depth estimation model for panoramic images. The main contributions are:

- A data curation pipeline that maps perspective images into spherical coordinates followed by image out-painting, substantially expanding the training data scale.
- A customized transformer with angle-based positional embeddings and a cross-attention design.
- A comprehensive evaluation of zero-shot and in-domain depth estimation across multiple benchmarks.

**Strengths:**

1. The paper clearly identifies limitations in current panoramic depth estimation methods and significantly expands the data scale, which is crucial for the field’s progress.
2. Benefiting from the enlarged dataset, the model outperforms existing methods on standard benchmarks.
3. It provides a comprehensive benchmark, including perspective-based models, which clarifies the current landscape.
4. The paper is clearly written and contains rich technical details.

**Weaknesses:**

1. Limited ablations on the dataset curation process.
   - Because ground-truth depth is only available in the central regions of the curated data (due to the limited FOV of perspective images; Fig. 3), it is unclear how missing annotations at the borders affect final performance.
   - The model uses image out-painting to compensate for limited FOV, but the visual quality of the out-painted regions is not clearly evaluated.

2. Limited novelty in some architectural components.
   The paper emphasizes angle-based positional embeddings (L269–299), cross-attention (L300–311), and a normal loss (Table 3, listed as a separate ablation item), but:
   - Positional embeddings: Improvements appear modest. In Table 3, removing $E_{\text{sphere}}$ yields only a small degradation (AbsRel 6.62 → 6.84) compared to gains from out-painting and the normal loss. It is also unclear whether the baseline here uses standard \(uv\) positional embeddings or no PE. The utility of the proposed PE seems smaller than claimed.
   - Cross-attention: The idea is not new and is under-ablated. Prior work (HUSH [1]) uses cross-attention to attend image features and SH features. Here, the method uses \(uv\) PE instead of SH and fixes image+feature as queries with PE as keys/values. Ablations should compare:
     (a) Cross-attention vs. self-attention (with image+PE as inputs, matching L303), and
     (b) Swapping queries/keys (PE vs. image features), following HUSH.
   - Normal loss: It contributes notably (Table 3), but normal losses have been used previously (e.g., [1]).

**Questions:**

1. Center vs. border performance: Can you provide ablations comparing performance in central regions versus border regions on panoramic images, to clarify benefits and limitations of the proposed dataset (related to Weakness 1-1)?
2. Out-painting quality: Can you include visual quality comparisons (e.g., SSIM, LPIPS, PSNR) between the out-painted images and corresponding ground-truth images (Weakness 1-2)?
3. Attention ablations: Please provide ablations for self-attention as well as query/key-value choices in cross-attention (Weakness 2).
4. Ablation dataset: Which evaluation dataset is used for the ablation results? The reported scores do not seem to align with any dataset in Table 2.

---

Reference
[1] HUSH: Holistic Panoramic 3D Scene Understanding using Spherical Harmonics, CVPR 2025.

---

> ### Author Response · Authors · 2025-11-20
> **Author's response to reviewer GN9K 1/5**
>
> **Response to weaknesses:**
>
> > **1.1** GT depth is only available in the central regions of the curated data, it is unclear how missing annotations at the borders affect final performance.
>
> We also kindly agree that the performance of DA$^{2}$ would surely be better if the depth label can also be accurately out-painted, and the missing depth label may decrease the performance especially near the poles. Therefore, we also incorporated a high-quality and native panoramic depth dataset (i.e., Structured3D) with a slightly higher sampling probability than curated perspective datasets (as shown in the *Supp*’s Tab. 4). And while training on this dataset, large errors and subsequently, large gradients near poles can be consistently observed, as illustrated in *Supp*’s Fig. 12, which forces the model to predict good distances on these areas.
>
> In order to better understand the missing annotations at borders (where the depth labels are missing) and centers (where the depth labels are available), we conducted 4 experiments using the native panoramic dataset (Structured3D): i. Using the entire dataset, with all depth labels available, noted as “All”. ii. Using the entire dataset, with all depth labels available in the first half and only “centers” depth labels are available in the second half, noted as “Half-center”. iii. Using the entire dataset, with all depth labels available in the first half and only “border” depth labels are available in the second half, noted as “Half-border”. iv. Using only the first half of the dataset, with all depth labels available, noted as “Half”.
>
> We originally expect that “All” is the best and “Half” is the worst due to the difference in scale.
> However, surprisingly, these 4 groups of experiments have basically the same results as the second row of Tab. 2 (only Structured3D, or S3D, is available, which corresponds to “All”). This phenomenon motivates us to more seriously consider the importance of the “diversity” of the “domains” in our training data. When the “quantity” is not the “the shortest plank in the barrel”, the importance of “diversity” is higher than our expectation. We thus re-analyze the scaling-law curves illustrated in Fig. 2 and Tab. 2. At the beginning, when the number of datasets used is small and the existing diversity is limited, performance grows very rapidly as additional datasets are introduced, since each new dataset contributes significantly to the diversity (e.g., different scene semantics, layout, lighting conditions, etc.). As illustrated, the introduction of Hypersim yields an improvement of 0.97 in AbsRel with only $\sim$33K's improvement in quantity, the ratio is 0.03 AbsRel(%)/K. The introduction of VKITTI further brings a clear improvement of 0.26 in AbsRel with $\sim$40K's improvement in quantity, but the ratio is only 0.007 AbsRel(\%)/K. As more datasets curated from different sources are introduced, the performance gained from both the diversity and quantity gradually becomes marginal. When the Dynamic Replica was introduced, the gain in AbsRel is only 0.04, though it brings an improvement of $\sim$290K in quantity.
>
> This discussion has been updated in PDF (*Supp*’s Sec. I).

---

> ### Author Response · Authors · 2025-11-20
> **Author's response to reviewer GN9K 2/5**
>
> > **1.2** The visual quality of the out-painted regions is not clearly evaluated.
>
> As an essential part of our panoramic data curation engine, the visual quality of the panoramic out-painter, i.e., the FLUX-I2P is better to be evaluated, which can help in understanding the quality of our curated data. Currently, there exist two well-recognized panoramic out-painters, Diffusion360 [1] and MVDiffusion [2].
>
> MVDiffusion is capable of generating 7 more perspective images based on a given input perspective image. Combined with the input, these 8 images (each has a horizontal FoV of 90$^\circ$) constitute a “partial” panorama, with each image separated by a horizontal gap of $45^\circ$. However, MVDiffusion assumes the input perspective image has a horizontal FoV of 90$^\circ$ and can not generate the upper and lower poles of the panorama, which is different from our requirement.
>
> Diffusion360's setting aligns with us closer. The inputs of both Diffusion360 and FLUX-I2P are a perspective image and its horizontal FoV (which corresponds to the out-painting mask as shown in *Supp*’s Fig. 13). And the output is the out-painted image. In our setting, the input perspective image will firstly be perspective-to-equirectangular (P2E) projected into the spherical space, and be consequently out-painted. During the out-painting, the P2E projected images, which perfectly correspond to the out-painting masks, are completely preserved. However, in Diffusion360's setting, the input perspective image may not be strictly preserved, as evident in our *Supp*’s Fig. 13.
>
> The qualitative comparisons of FLUX-I2P with Diffusion360 and MVDiffusion is shown in *Supp*’s Fig. 13. As illustrated, FLUX-I2P is able to generate panoramas with superior visual quality, high realism, and strong semantic coherence with the input image. While Diffusion360 and MVDiffusion may exhibit clear inconsistencies between their out-painted content and the input image (e.g., Diffusion360's result on the left case, MVDiffusion's result on the right case), exhibit oversaturated style (e.g., Diffusion360's result on the right case).
>
> In order to quantitatively evaluate our FLUX-I2P with SoTA baselines, we conduct a user study across a wide range of evaluation aspects, as shown in *Supp*’s Tab. 6 (or the table below). This is because there rarely exists a well-recognized benchmark for panoramic image out-painting, where we can calculate metrics like SSIM, LPIPS, PSNR; and it may not very straightforward to evaluate “generation abilities” using these reconstruction metrics. Also, considering the in-complete panoramic out-painting of MVDiffusion, which largely differs from the settings of FLUX-I2P and Diffusion360, we kindly remove MVDiffusion for the comparison. Four metrics are considered in this quantitative evaluation:
> i. The overall visual quality of the panorama (an ERP image) (VQ-ERP).
> ii. The semantic consistency of the out-painted content and the input image (IN-OUT-CON).
> iii. The circular seams (SEAM).
> iv. The visual quality of the panorama under immersive perspective projection (VQ-PER).
>
> Table: Winning rate of FLUX-I2P over Diffusion360 [1] measured on 30 cases by human preferences (the higher, the better). A total of 5 people participated (no authors) and 150 comparisons were made. FLUX-I2P consistently outperforms Diffusion360 in the above evaluation perspectives. Please see *Supp*’s Fig. 14 for the interface of the user study example.
>
> | Competition | VQ-ERP | IN-OUT-CON | SEAM | VQ-PER |
> |:--|:--:|:--:|:--:|:--:|
> | FLUX-I2P (Ours) vs. Diffusion360 | 72.7 % | 92.7 % | 55.3 % | 79.3 % |
>
> This discussion has been updated in PDF (*Supp*’s Sec. J).
>
> [1] Mengyang Feng, et al. Diffusion360: Seamless 360 Degree Panoramic Image Generation based on Diffusion Models. arXiv 2023.
>
> [2] Shitao Tang, et al. MVDiffusion: Enabling Holistic Multi-view Image Generation with Correspondence-Aware Diffusion. NeurIPS 2023.

---

> ### Author Response · Authors · 2025-11-20
> **Author's response to reviewer GN9K 3/5**
>
> > **2.1** Positional embeddings: It is also unclear whether the baseline here uses standard (uv) positional embeddings or no PE. The utility of the proposed PE seems smaller than claimed.
>
> We thank you for your interest in SphereViT and your in-depth reading. The baseline ViT model does use standard 2D sine-cosine positional embeddings based on the rectangular pixel coordinates (u, v), consistent with conventional ViTs. In contrast, our SphereViT proposes “sphere-aware” spherical embeddings ($\phi, \theta$) derived from the true azimuth and polar angles of each pixel.
> This change explicitly encodes the non-uniform sampling of the ERP images and the spherical structure of the $360^\circ$ domain; these are key properties that (u, v) embeddings cannot model.
> While the overall AbsRel gain may appear moderate, we emphasize that this is because:
>
> i. The introduction of spherical embeddings brings reduced curvature distortion (e.g., more straight walls in paper’s Fig. 6 (a)) and improved geometric stability primarily near the poles where the spherical distortions are significant, which may not be fully captured by globally averaged metrics such as AbsRel or RMSE.
> ii. The panoramas in real-world benchmarks are usually in-complete also primarily near the poles. As shown in *Supp*’s Fig. 15.
>
> This discussion has been updated in PDF (*Supp*’s Sec. M.2).
>
> > **2.2** Discussions with HUSH [3] regarding the cross-attention, and more ablations (cross-attention vs. self-attention, swapping Q and K & V in cross-attention).
>
> **Discussions with HUSH [3] regarding the cross-attention**
>
> HUSH [3] uses spherical harmonic (SH) basis functions as spherical priors aiming for better feature learning on spherical structures. The core modules are: i. SH-based hierarchical attention module, and ii. SH basis index module. In the first module, SH basis functions serve as geometric priors to guide hierarchical cross-attention between spherical geometry and image features. The second module selects important indexes of SH bases based on the similarity of the task specific feature $f_T$ and embedded SH basis $B_n$, to adaptively emphasize relevant SH bases.
>
> Both HUSH and SphereViT use cross-attention, but the ways how they are using them are different. SphereViT uses spherical embeddings $E_\text{sphere}$ as keys and values, and the image feature as queries. While HUSH, in its SH-based hierarchical attention module, initially uses SH feature as query and image feature $f_1$ as key and value. Then, HUSH sequentially and alternatively uses image feature (e.g., $f_2, f_3$) or SH feature as query, the output of last cross-attention acts as key and value. HUSH only uses the initial image feature $f_1$ as the key and value, and SH feature as query.
>
> The underlying reason for such differences in injecting spherical awareness is that the structural prior in HUSH (SH coefficients) is learnable but in SphereViT the prior is fixed. SH bases are base functions defined on spherical space. They don’t directly represent the spherical structure itself, but are more likely a defined feature space which should also be refined (or learned) with the image features. HUSH also selects different SH bases to accommodate different tasks. While in SphereViT, the spherical embeddings $E_\text{sphere}$ directly represent the spherical structure, and thus can be fixed as a global context and be shared across different panoramas. This is similar to the way diffusion models [4] inject users' prompts as global context during image generation.
>
> We sincerely hope our analysis above can help in addressing your concerns regarding the architecture of SphereViT. And we also follow your suggestions in conducting more ablation experiments, please see our responses below.
>
> This discussion has been updated in PDF (*Supp*’s Sec. M.1).

---

> ### Author Response · Authors · 2025-11-20
> **Author's response to reviewer GN9K 4/5**
>
> **More analysis about our cross-attention design.**
>
> In a cross-attention layer, the computation is defined as: $\texttt{CrossAttn}\left(Q,K,V\right)=\operatorname{SoftMax}\left(\frac{Q W_Q\left(K W_K\right)^{\top}}{\sqrt{D_k}}\right)\left(V W_V\right),$
>
> Firstly, the output has the same spatial and temporal dimension as query (Q), that is, the attention's output corresponds strictly with query, where each token incorporates information from key (K) and value (V). This indicates K and V provides “reference information sources” and Q is the “target feature” being updated. Secondly, considering the information flow, the Q determines “where to look”, its features generate attention weights with K that decide which parts of V are relevant. The attended information is then aggregated back to Q’s positions. Consequently, the data path updates Q while K and V act as conditioning sources that guide the update. Then, during backpropagation the dominant gradient flows toward Q’s parameters, since the loss is computed on the attention's output that shares Q’s shape. The model thus learns to adjust Q’s representations so that they can better extract and integrate information from K,V.
>
> Thus, the Q defines the representation to be enhanced, while the K, V provide context or prior knowledge. This principle underlies SphereViT’s design, where image features (as Q) are refined through attention to the spherical embedding (as K, V), enabling the model to absorb geometric priors without modifying the fixed spherical coordinates themselves. Similar design has been well recognized in Stable Diffusion [4], where cross-attention is used to inject users' prompts as global context. During image generation, the image latent features (as Q) are gradually refined through attention to the ``fixed'' prompt embedding (as K, V), enabling the model to generate images that accurately respond to users' preferences.
>
> We hereby kindly revise our statement in line 269-277. Our spherical embedding $E_\text{sphere}$ is only conceptually inspired from the u, v positional embeddings in standard ViT, and inspired from Stable Diffusion[4] in implementation. That is because $E_\text{sphere}$’s role is not merely to assign each visual token a different position as in standard ViT, but to inject spherical awareness (a known structural prior) into the image feature, which is similar to Stable Diffusion that inject users’ prompt into image generation.
>
> This discussion has been updated in PDF (*Supp*’s Sec. N).
>
> **More ablations (cross-attention vs. self-attention, swapping Q and K & V in cross-attention)**
>
> When the cross-attention module is replaced by self-attention after the addition of the spherical embedding and image features, the model performance also drops notably, as shown in *Supp*’s Tab. 7 (or the table below). This is because self-attention treats the spherical embedding ``part of'' the feature channels, without explicitly establishing any directional interaction between the image features and the geometric prior. Consequently, the model fails to effectively incorporate spherical structural information, leading to weaker distortion awareness and reduced overall accuracy.
>
> Swapping query and key / value in Cross-attention fundamentally changes the direction of ``attention flow''. As analyzed earlier, in a cross-attention layer, the tokens in Q are the ones being refined, while the K / V tokens act as reference sources. When the roles are swapped, the spherical embedding becomes the Q, the attention tries to update the spherical geometry considering the image features. This disrupts semantic alignment between image features and spherical geometry, preventing the model from absorbing the spherical priors. As a result, the model fails to converge and produces severely degraded performance, confirming that the original attending direction is essential for effective spherical conditioning.
>
> Table: Ablation studies about SphereViT's architectural design. The **best** and _second best_ performances are bolded and italicized. Following paper’s Tab. 2 and 3, the averaged results across multiple benchmark datasets are reported. Unit is percentage (%). “/” means the model doesn’t converge.
>
> | Operations for ablation | AbsRel ↓ | RMSE ↓ | δ₁ ↑ | δ₂ ↑ |
> |:--|:--:|:--:|:--:|:--:|
> | Swap Q and K / V in cross-attention | / |/ |/ |/ |
> | Replace cross-attention by concatenation and self-attention | 7.42 |23.04 | 94.31 | 97.98 |
> | Replace cross-attention with self-attention | _7.29_ | _22.45_ | _94.60_ | _98.13_ |
> | **Nothing changed** | **6.62** | **20.63** | **95.73** | **98.51** |
>
> This discussion has been updated in PDF (*Supp*’s Sec. N, N.1, and N.3).
>
> [3] Jongsung Lee, et al. HUSH: Holistic Panoramic 3D Scene Understanding using Spherical Harmonics. CVPR 2025.
>
> [4] Robin Rombach, et al. High-Resolution Image Synthesis with Latent Diffusion Models. CVPR 2022.

---

> ### Author Response · Authors · 2025-11-20
> **Author's response to reviewer GN9K 5/5**
>
> > **2.3** Discussions with HUSH [3] regarding the normal losses
>
> HUSH states in its abstract: “Finally, by combining the scene features with task-relevant features in the task-specific heads, we perform various scene understanding tasks, including depth, surface normal and room layout estimation.”. While we focus on zero-shot panoramic depth estimation, because depth estimation can be considered as a feed-forward 3D point cloud reconstruction, which enables a wide range of applications compared to only the 2D visual content.
>
> Both HUSH and SphereViT use normal loss, but the motivations are different. HUSH “performs various scene understanding tasks” (i.e., depth, normal, layout) and its predictions of different tasks come from different prediction heads and different SH bases. Thus, the normal loss (also the depth loss, gradient loss) are the main losses that should be included during training because they directly correspond to the claimed tasks. While SphereViT focuses on depth only, and we consider depth as an important foundational 3D task. The motivation why we use normal loss is aiming to mitigate the artifacts of the reconstructed 3D point cloud (please see our paper’s Fig. 6) especially near the poles. These artifacted areas generally share similar depth values with GT, yet their normal values differ from GT markedly. Thus, the normal loss of SphereViT is an important auxiliary loss to improve the quality of the predicted depth, and the reconstructed 3D point cloud.
>
> This discussion has been updated in PDF (*Supp*’s Sec. M.3).
>
> **Response to questions:**
>
> Please see point 1.1, 1.2, 2.1, 2.2 in our responses to weaknesses for your questions 1, 2, 3. Also please see our *Supp*’s Sec. I, J, M.1, M.2, M.2, N, N.1, N.3.
>
> > **4.** Which evaluation dataset is used for the ablation results?
>
> The reported results in paper’s Tab. 2 and 3 are the averaged results across multiple benchmark datasets used in Tab. 1, as stated in line 379 of the paper.

---

### Official Review · Reviewer_t7tq · 2025-11-01

**Soundness:** 3
**Presentation:** 3
**Contribution:** 3
**Rating:** 6
**Confidence:** 3

**Summary:**

DA2 trains an end-to-end panoramic depth (scale-invariant distance) model by first,  curating a large dataset containing panoramas from perspective RGB-D via P2E projection plus generative panoramic out-painting (FLUX-I2P), and develop a sphereViT backbone that cross-attends from image features to a fixed spherical angle embedding. On benchmarks, DA2 reports the best zero-shot results, beating strong zero-shot baselines like UniK3D.  Authors highlight better or comparable speed to end-to-end competitors while much faster than fusion pipelines.

**Strengths:**

1. Strong empirical gains. The proposed method demonstrates strong performance compared to both zero-shot fusion, zero-shot end-to-end and in-domain methods.
2. The proposed method scales well. The paper shows clear scaling-law-like improvements as more perspective data are "panoramaized"
3.  The proposed architecture is effective. The SphereVIT's fixed spherical embedding plus cross attention is a simple but effective way to inject spherical awareness, which shows strong results.

**Weaknesses:**

1. It seems there is a supervision mismatch introduced by the generative out-painting. The curation engine out paints RGB to a full panorama but supervised only the P2E covered part of depth (no out-painted depth due to acc concern). Hence, many pixels in the training panoramas lack ground-truth depth, yet their RGB content is model-generated. This opens risks of learned correlation with out-painter priors and biased geometry in unsupervised regions (especailly near poles and seams).  The paper shows gains with out-painting but it doesn't quantify how much of the test-time improvement stems from distribution mathcin got FLUX-I2P artifacts compared with real geometric learning.

2. Injecting coordinates/angles and using attention to a fixed positional bank is not conceptually new. [1] mitigates ERP distortion via spherical tangent tokens and transformer design. [2] operates directly on spherical meshes, and several recent spherical ViTs or positional-encoding approaches aim at similar goals.

3. Metric depth is not addressed when comparing to Unik3D. DA2 predicts scale-invariant distance, not metric depth. With Unik3D showing metric 3d across cameras (including panoramic), it is important to discuss scale recovery or to provide an optional scale head, since it is important for downstream applications like AR or robotics.


[1] Shen, Zhijie, et al. "PanoFormer: panorama transformer for indoor 360∘ depth estimation." European Conference on Computer Vision. Cham: Springer Nature Switzerland, 2022.
[2] Yan, Qingsong, et al. "Spheredepth: Panorama depth estimation from spherical domain." 2022 International Conference on 3D Vision (3DV). IEEE, 2022.

**Questions:**

1. What fraction of pixels per training panorama actually have GT depth supervision, on average and by latitude?

2. Can we consider add a simple scale-head (or post-hoc scale regressor) and evaluating metric errors where ground truth exists. Compare to UniK3D on the same splits.

---

> ### Author Response · Authors · 2025-11-20
> **Author's response to reviewer t7tq 1/2**
>
> **Response to weaknesses:**
>
> > **1.** It seems there is a supervision mismatch introduced by the generative out-painting.
>
> In the authors' understanding, this concern has two parts: One is about the distribution mismatch between the panoramic out-painter and the testing scenarios (including experiments and potential applications). The other one is about the in-complete depth supervision, which may “cause biased geometry especially near the poles and seams”. Please see our responses on these two parts in the below two paragraphs:
>
> About the first part, we also consider the distribution mismatch between training and testing (applications, experiments) as a critical issue that may affect the final zero-shot performance. And we think making the training data more diverse and comprehensive is an effective way to mitigate the distribution mismatch. Thus, we comprehensively select multiple high-quality perspective datasets in different domains (i.e., captured or rendered in different scenes with different styles, etc.) as the sources of our panoramic data curation engine. Some examples of the perspective data sources and the curated data are shown in our *Supp*’s Fig. 11. Also, the training data of the out-painter is also diversely combined from commercially purchased data and self rendered data, in order to fit the diverse domains of multiple perspective data sources. Some examples of the out-painter's training data are shown in *Supp*’s Fig. 10.
>
> About the second, we also agree that the performance of DA$^{2}$ would surely be better if the depth label can also be accurately out-painted, and the missing depth label may decrease the performance especially near the poles. Thus, we also incorporated a high-quality and native panoramic depth dataset (i.e., Structured3D) with a slightly higher sampling probability than each panoramically transformed perspective dataset (as shown in Tab. 4 in *Supp*’s Sec. B). And while training on this dataset, larger gradients near the poles can be observed (as illustrated in *Supp*’s Fig. 12), which forces the model to predict good distances on these areas.
>
> This discussion has been updated in PDF (*Supp*’s Sec. G and I).
>
> > **2.** Injecting coordinates/angles and using attention to a fixed positional bank is not conceptually new. PanoFormer and SphereDepth proposed a potentially similar architectural design with SphereViT.
>
> PanoFormer[1] proposed a STLM to initialize the related token positions (replacing conventional ViT pos. emb.), using the spatial transformations among a. ERP domain, b. spherical domain, c. tangent domain. Then, PanoFormer proposed a token flow to “learn” a bias of STLM, to refine it based on image features. SphereDepth[2] leverages ISM to represent the spherical structure and use interpolation to transfer the ERP image pixels into (and back from) each sample point on the ISM (controlled by both MR and TR).
>
> Both SphereViT and PanoFormer propose a “sphere-aware” embedding to replace the conventional one. But compared with PanoFormer, SphereViT directly and explicitly inject the entire $360^\circ\times 180^\circ$ spherical structure into the network, while PanoFormer’s STLM focuses on the relative positions within a tangent projection patch, which is like a “fine-grained” version of cubemaps.
>
> Both SphereDepth and SphereViT directly and explicitly inject spherical structure into the network. SphereDepth does this even more explicitly as it directly designs sphere network blocks and both the input and prediction are spherical representations sampled on ISM. However, as stated in their original paper: “representing a high-resolution panorama image needs a spherical mesh with high resolution (i.e., high MR) and more triangles.”, and  “The topological complexity of the spherical mesh exponentially grows with the number of triangles (i.e., MR) increasing, which degrades the computing efficiency and increases the memory footprint.”. Also, interpolation is needed to project dense pixels into and back from sampled points on ISM. Thus, compared to SphereDepth, SphereViT is more streamlined and may have higher efficiency.
>
> This discussion has been updated in PDF (*Supp*’s Sec. K).
>
> [1] Shijie Shen, et al. PanoFormer: Panorama Transformer for Indoor 360 Depth Estimation. ECCV 2022.
>
> [2] Qingsong Yan, et al. SphereDepth: Panorama Depth Estimation from Spherical Domain. 3DV 2022.

---

> ### Author Response · Authors · 2025-11-20
> **Author's response to reviewer t7tq 2/2**
>
> > **3.** Metric depth is not addressed when being compared to Unik3D.
>
> We sincerely agree that metric depth estimation can expand the downstream applications. The current version of this work only supports “scale-invariant” or “biased” because the authors consider the “metric-scale” a little ill-posed because sometimes the metric-scale can’t be reflected in the panoramas. For example, if the entire scene is scaled equal-proportionally and linearly, the 360$^\circ$ pictures captured will still remain the same. However, these situations are rare. Thus, we also train a metric version of DA$^2$ by directly supervising the model with metric GT depth, to explore its metric depth performance. As illustrated in *Supp*’s Tab. 9 (also in below), thanks to the data curation engine and the SphereViT which is ``sphere-aware'', DA$^2$ consistently achieves higher performance compared with UniK3D.
>
> Table: Quantitative comparisons on metric depth estimation, compared with UniK3D. The **best** and _second best_ performances are bolded and italicized.
> | Method | Stanford2D3D |     |     |     | Matterport3D |     |     |     | PanoSUNCG |     |     |     |
> |:--|:--:|:--:|:--:|:--:|:--:|:--:|:--:|:--:|:--:|:--:|:--:|:--:|
> |  | AbsRel ↓ | RMSE ↓ | δ₁ ↑ | δ₂ ↑ | AbsRel ↓ | RMSE ↓ | δ₁ ↑ | δ₂ ↑ | AbsRel ↓ | RMSE ↓ | δ₁ ↑ | δ₂ ↑ |
> | UniK3D | _17.95_ | _48.50_ | _78.23_ | _86.63_ | _22.24_ | _66.80_ | _66.34_ | _92.81_ | _12.60_ | _27.20_ | _88.90_ | _98.47_ |
> | DA² (Ours) | **16.56** | **45.73** | **80.04** | **90.32** | **17.53** | **60.85** | **68.21** | **96.13** | **11.23** | **21.03** | **89.20** | **98.53** |
>
> This discussion has been updated in PDF (*Supp*’s Sec. O).
>
> **Response to questions:**
>
> > **1.** What fraction of pixels per training panorama actually have GT depth supervision, on average and by latitude?
>
> On average, most perspective data samples we used have an X FoV of $70^\circ\sim90^\circ$ and the height usually equals the width. Thus, the fraction of GT depth labels in the curated panoramas is: $80/180 * 80/360 \approx 9.9$%. By latitude, the GT depth labels range from $-40^\circ$ to $40^\circ$. Moreover, though the fraction in area is only 10%, the fraction in semantic is much larger, because the ERP images contain much richer semantics near the equator than the poles.
>
> This discussion has been updated in PDF (*Supp*’s Sec. B, as the second paragraph).
>
> > **2.** Can we consider adding a simple scale-head (or post-hoc scale regressor) and evaluating metric errors where ground truth exists. Compare to UniK3D on the same splits.
>
> Yes. We also trained a metric version of DA$^2$. Please see point 3 in our responses to your weaknesses and *Supp*’s Sec. O for more details.

---

> > ### Comment · Reviewer_t7tq · 2025-11-26
> >
> > Thanks to the authors for the rebuttal! It solved my concerns.

---

### Author Response · Authors · 2025-11-20
**Author's general response to AC and all reviewers**

We sincerely thank all reviewers for their time and insightful, professional comments on our submission. We also sincerely thank our AC and SAC for their time in coordinating the discussion process. Thank you reviewers, AC, and SAC for your effort in pushing forward ICLR’26!

Firstly, we are highly encouraged and grateful to see the consistently positive ratings from all reviewers, and the strengths of our submission mentioned by our reviewers after their careful reading, such as:

1. The paper “clearly identifies current limitations” (GN9K), targets on “critical issues” (RcRx), and “represents a step toward universal depth perception across all directions” (c7zZ).
2. The data curation engine “significantly expands the data scale, which is crucial for the field’s progress” (GN9K, xeLR, RcRx), and “the model scales well” with “clear scaling-laws” (t7tq, xeLR)
3. The SphereViT “introduces a neat and elegant solution to the problem of spherical distortion” (xeLR), is “sphere-aware” (c7zZ), and “effectively injects spherical awareness” (t7tq, RcRx).
4. The paper “provides a comprehensive benchmark” (GN9K, c7zZ, xeLR, RcRx), and shows “strong empirical gains” (t7tq, RcRx).
5. “The paper is clearly written” (GN9K, RcRx), “easy to follow” (xeLR), and “contains rich technical details” (GN9K).
6. “The release of the code and the large-scale curated panoramic dataset is a big contribution” (xeLR, RcRx), and “will be a substantial asset to the research community” (xeLR).

And the good (t7tq, c7zZ, xeLR, RcRx) soundness,  good (t7tq, xeLR, RcRx) and excellent (GN9K, c7zZ) presentation, good (t7tq, GN9K, xeLR, RcRx) contribution.

We are also highly grateful to see the weaknesses pointed out and the questions raised by the reviewers. These are valuable suggestions from different aspects to improve the overall quality of this paper. We carefully and responsibly replied to each reviewer in a strictly point-by-point manner, aiming to settle all concerns from each reviewer by providing additional discussions, doing and reporting new experiments, etc. We also updated the PDF submission (paimarily in *Supp*'s Sec. G$\sim$Q) according to reviewers’ comments and our responses. The key improvements are summarized below:


**Training data and distribution**
1. Discussions about the Potential Distribution Mismatch Between the Out-painter and Testing Scenarios (Please see **Sec. G**. In response to: 7tq, c7zZ, RcRx).
2. Discussions about Real-world Performance as the Training Data are Mostly Synthetic (Please see **Sec. H**. In response to: 7tq, c7zZ, RcRx).

**The panoramic data curation engine**
1. Discussions about In-complete Depth Supervision (Please see **Sec. I**. In response to: GN9K, xeLR, RcRx).
2. The Visual Quality of Out-painted Regions (Please see **Sec. J**. In response to: GN9K, xeLR)

**The architectural design of SphereViT**
1. Discussions about SphereViT with PanoFormer and SphereDepth (Please see **Sec. K**. In response to: t7tq).
2. Discussions about SphereViT with S$^2$Net (Please see **Sec. L**. In response to: xeLR).
3. Discussions about SphereViT with HUSH in terms of the Spherical Embedding; Cross-attention; and Normal Losses (Please see **Sec. M**. In response to: GN9K)
4. More Ablations about SphereViT's Architectural Design, e.g., Cross-attention vs. Addition and Self-attention; Cross-attention vs. Concatenation and Self-attention; Swapping K/V and Q in Cross-attention; etc. (Please see **Sec. N**. In response to: GN9K, xeLR)

**Training a metric depth version of DA$^2$.** (The current is scale-invariant. Please see Sec. D for the difference among: Metric & Scale-invariant (Biased) & Affine-invariant (Relative))
1. Metric Depth Version of DA$^2$ (Please see **Sec. O**. In response to: t7tq)

**About the left-right seams**
1. Discussions about the Left-right Seams and Potential Solutions (Please see **Sec. P**. In response to: c7zZ, xeLR, RcRx).

**More Qualitative Results**
1. More Qualitative Results Compared with UniK3D and MoGev2 (Please see **Sec. Q**. In response to: RcRx).

We are very grateful for the prompt responses from Reviewer t7tq, xeLR, and RcRx confirming that most / all of his / her concerns have been resolved. And we sincerely hope that our rebuttal has addressed the concerns from Reviewer GN9K and c7zZ as much as possible. Also, please note that the summary listed above doesn’t represent all the rebuttal and revisions of this submission. Some minor discussions between the reviewers and authors are not listed, e.g., real-world data capture, applications, typos, etc.

Thanks again to all reviewers and AC, SAC for your valuable time and your effort in pushing forward ICLR’26!

Thanks and best regards,

Authors of submission #190

---

### Meta-Review · Area_Chair_7xRK · 2026-01-06

**Summary:**

Overall, the reviewers were initially positive (four of them rated 6s and one 8), but they shared concerns regarding the synthetic data generation pipeline and the architectural novelty. The authors provided a very strong rebuttal, including running new experiments, adding ablations, and clarifying comparisons, which I believe have convinced reviewers to support acceptance. Specifically, there are a few key concerns shared by multiple reviewers.

**1. Data Curation & Supervision Mismatch**
The authors incorporated native panoramic datasets to force gradients in missing areas. They also added a user study comparing FLUX to other out-painters to prove quality, and also provided some analysis on the domain gap. The answers have adequately addressed the concern.

**2. Architectural Novelty**
Reviewers pointed out similarities to a couple of existing works, and the authors made clarifications to each of the work, and also added some additional ablations.

**3. Seam Artifacts**
Reviewers c7zZ, xeLR, RcRx all mentioned the noticeable seams at the boundary in the point cloud visualization. The authors admitted the limitation, and proposed a horizontal rotation augmentation strategy in the rebuttal to mitigate it, which reviewers seemed to accept as a reasonable solution.

I think the authors successfully demonstrated the effectiveness of the proposed method, and have addressed most comments during rebuttal. Therefore, I recommend accepting the paper.

**Reviewer Concerns:**

All reviewers' concerns have been addressed. 3 out of 5 reviewers (t7tq, xeLR, RcRx) explicitly said that their concerns have been addressed, and they keep their positive ratings. For 2 other reviews, after checking them, I believe they have been addressed.

**Reviewer Scores:**

**t7tq**: The reviewer indicates that the rebuttal has solved their concerns. I think the reviewer will likely keep the original rating 6 or raise it to 8.

**GN9K**: The reviewer has a detailed list of good questions, and after checking the authors' answers, I believe they have adequately addressed them all. Therefore, I believe the reviewer will likely either keep the original rating 6 or raise it to 8.

**c7zZ**: Probably will just keep the original rating 6 or raise it to 8.

**xeLR**: The reviewer indicates that the rebuttal has solved their concerns and maintain the original positive rating (6). Therefore, no rating change

**RcRx**: The reviewer indicates that the rebuttal has addressed their concerns. Therefore, the reviewer will probably keep the good rating (8).

---

### Decision · Program_Chairs · 2026-01-26

Accept (Poster)